# Time-sensitive prefrontal involvement in associating confidence with task performance illustrates metacognitive introspection in monkeys

Yudian Cai [1,2,3], Zhiyong Jin[1,2,3], Chenxi Zhai [1], Huimin Wang[1,4,5], Jijun Wang[6,7,8], Yingying Tang [8✉] & Sze Chai Kwok [1,2,3,5✉]

Metacognition refers to the ability to be aware of one's own cognition. Ample evidence indicates that metacognition in the human primate is highly dissociable from cognition, specialized across domains, and subserved by distinct neural substrates. However, these aspects remain relatively understudied in macaque monkeys. In the present study, we investigated the functionality of macaque metacognition by combining a confidence proxy, hierarchical Bayesian meta-d' computational modelling, and a single-pulse transcranial magnetic stimulation technique. We found that Brodmann area 46d (BA46d) played a critical role in supporting metacognition independent of task performance; we also found that the critical role of this region in meta-calculation was time-sensitive. Additionally, we report that macaque metacognition is highly domain-specific with respect to memory and perception decisions. These findings carry implications for our understanding of metacognitive introspection within the primate lineage.

[1] Shanghai Key Laboratory of Brain Functional Genomics, Key Laboratory of Brain Functional Genomics Ministry of Education, Shanghai Key Laboratory of Magnetic Resonance, Affiliated Mental Health Center (ECNU), School of Psychology and Cognitive Science, East China Normal University, Shanghai 200062, China. [2] Division of Natural and Applied Sciences, Duke Kunshan University, Kunshan, Jiangsu 215316, China. [3] State Key Laboratory of Cognitive Neuroscience and Learning, Beijing Normal University, Beijing 100875, China. [4] NYU-ECNU Institute of Brain and Cognitive Science at NYU Shanghai, Shanghai 200062, China. [5] Shanghai Changning Mental Health Center, Shanghai 200335, China. [6] Brain Science and Technology Research Center,  Shanghai Jiao Tong University, Shanghai 200030, China. [7] CAS Center for Excellence in Brain Science and Intelligence Technology (CEBSIT), Chinese Academy of Science, Shanghai 200031, China. [8] Shanghai Key Laboratory of Psychotic Disorders, Shanghai Mental Health Center, Shanghai Jiao Tong University School of Medicine, Shanghai 200030, China. ✉email: yytang0522@gmail.com; sze-chai.kwok@st-hughs.oxon.org

Metacognition, the ability to monitor and evaluate one's own cognitive processes, is believed to be unique to humans. Ample evidence indicates that neural underpinnings supporting metacognitive abilities are different from cognitive processes[1–9]. A number of human transcranial magnetic stimulation (TMS) studies have implicated the dorsolateral prefrontal cortex (dlPFC) in meta-perceptual judgements more than in perceptual judgements[10–12]. This evidence indicates that the prefrontal cortex, especially the lateral prefrontal cortex (lPFC), is a key region in the metacognitive mechanism[8,13,14].

Less understood, however, is whether the importance of dlPFC is conserved in other species, such as nonhuman primates. Only one extant study has investigated the role of macaques' dlPFC in meta-perceptual processes. That study found that in a visual-oculomotor task, single neurons in the dlPFC encode metacognitive components of decision-making[15]. Based on a review of the literature in both human and NHP studies, we believe that the dlPFC could likely act as a key site for (perceptual) introspection in the macaques. Apart from meta-perceptual processes, neural activations in dorsal PFC and anterior PFC in the macaque brain are associated with metacognition of experienced object recognition[16,17]. We sought to expand on the findings of those studies; our first aim was to test for any functional role of the monkey dlPFC in meta-perception independent of perception itself. To achieve this goal, we applied single-pulse transcranial magnetic stimulation to the dlPFC (BA46d) of monkeys while they performed a perceptual resolution judgement task. We adopted a temporal wagering paradigm to measure the animals' trial confidence in each trial[18–20]. Following each perceptual decision, the animals were required to wait for an unknown and variable period by keeping their hand on the screen before they qualified for any possible reward. The amount of time wagered on their decision in a given trial was used as a proxy for confidence in the decision.

Taking advantage of single-pulse TMS, we intended to ascertain the precise window in which meta-computation is carried out. An electrophysiology study reported that information carried by lateral intraparietal cortex (LIP) neurons at the time of decision is sufficient for predicting subsequent confidence-related neural responses[21]. However, single-pulse TMS of the dorsal premotor cortex (PMd) impairs confidence reports in both the pre-response and post-response windows[22], suggesting that late-stage evidence accumulation might also be required for metacognitive processes. To more precisely determine the critical phase in which meta-calculation takes place, we included two time-sensitive TMS conditions: on-judgement and on-wagering stimulation. Specifically, we applied TMS either 100 ms after stimulus onset (on-judgement phase) or 100 ms after the animal's decision (on-wagering phase). If the critical phase of meta-calculation was within the decision stage, we would expect metacognition deficits when TMS was applied during the on-judgement phase. In contrast, if the meta-computation was at a later stage (e.g., concurrent with processes associated with wagering), we would expect metacognition deficits when TMS was applied during the on-wagering phase.

There is evidence that efficient metacognition in one task can predict good metacognition in another task[23–28]. For example, monkeys' ability to transfer their metacognitive judgement from a perceptual test to a memory test shows that they can employ domain-general signals to monitor the status of cognitive processes and knowledge levels[29,30], suggesting that metacognition is generalized across domains. However, mounting anatomical[3,31], functional[6], and neuropsychological[4,32,33] evidence in the human research literature increasingly points to the domain specificity of metacognition, indicating that humans possess specialized metacognitive abilities for different domains[6,23,33–35]. Here, we posed the question of whether macaques show domain-specific components of metacognition[29]. To this end, we trained two additional monkeys to perform a temporal-memory task in combination with the wagering task. Making use of the data collected in both experiments, we assessed both the covariation and the divergence between metacognitive abilities in the two domains.

## Results

In the following, we will report results obtained from four monkeys who participated in two distinct tasks tapping into two metacognitive domains. Most critically, we measured the animal's trial-wise confidence level using a time-wagering paradigm.

**Metacognition in monkeys in both the memory and perception domains**. To show that macaques are capable of metacognition, we quantified this capacity using bias-free metacognitive efficiency (H-model $meta\text{-}d'/d'$). We compared animals' scores to zero using one-sample $t$ tests and found that the meta-index values of all monkeys were above zero for both tasks (Fig. 1c, d; meta-perception: H-model $meta\text{-}d'/d'$: Mars, $t_{(19)} = 5.685$, $p < 0.001$; Saturn, $t_{(19)} = 5.639$, $p < 0.001$; Uranus: $t_{(19)} = 10.55$, $p < 0.001$; Neptune, $t_{(19)} = 9.458$, $p < 0.001$; meta-memory: H-model $meta\text{-}d'/d'$: Mars, $t_{(19)} = 9.012$, $p < 0.001$; Saturn, $t_{(19)} = 5.639$, $p < 0.001$; Uranus: $t_{(19)} = 4.159$, $p < 0.001$; Neptune, $t_{(19)} = 3.621$, $p < 0.001$).

We then replicated the results with the phi coefficient (meta-perception: phi coefficient: Mars, $t_{(19)} = 3.643$, $p < 0.001$; Saturn, $t_{(19)} = 6.245$, $p < 0.001$; Uranus: $t_{(19)} = 6.722$, $p < 0.001$; Neptune, $t_{(19)} = 3.423$, $p < 0.001$; meta-memory: phi coefficient: Mars, $t_{(19)} = 4.135$, $p < 0.001$; Saturn, $t_{(19)} = 2.962$, $p = 0.004$; Uranus: $t_{(19)} = 2.252$, $p = 0.018$; Neptune, $t_{(19)} = 1.838$, $p = 0.041$). We further confirmed the reliability between Phi and Hmodel-meta $d'/d'$. We found the two metrics were highly correlated (Pearson correlation: experiment domain-comparison, $r = 0.7916$, $p < 0.001$; experiment TMS, $r = 0.7415$, $p < 0.001$; Fig. 1e, f).

To further validate these results, we combined all trials per monkey across all days and then performed subject-based distribution simulations on each monkey. By randomly shuffling all the pairings between responses (correct/incorrect) and their corresponding confidence levels (high/low) within each subject, we generated 2000 random pairings for each animal and simulated 4000 metacognitive scores per animal (both the H-model $meta\text{-}d'/d'$ and the phi coefficient). These scores represent cases in which the animals had no metacognitive ability. We then tested these simulated scores against animals' actual scores using a minimum statistic method;[36] we found that the animals indeed performed significantly above chance metacognitive ability in both tasks (all $p$ values <0.001; Table 1).

As a control to rule out any possible contribution of training effects, we compared the animals' metacognition scores between the first ten days and the second ten days of testing. We found no difference between the first ten days and the second ten days of metacognitive performance in either perception (H-model $meta\text{-}d'/d'$): ($t_{(39)} = -0.314$, $p = 0.755$) or memory (H-model $meta\text{-}d'/d'$): ($t_{(39)} = 0.89$, $p = 0.378$). These results show that the metacognitive ability of the animals was stable across the whole testing period. For completeness, we checked the monkeys' cognitive performance and found that they improved moderately in the second half in the memory task (accuracy: $t_{(39)} = -2.266$, $p = 0.029$) but not in the perception task ($t_{(39)} = -1.083$, $p = 0.285$).

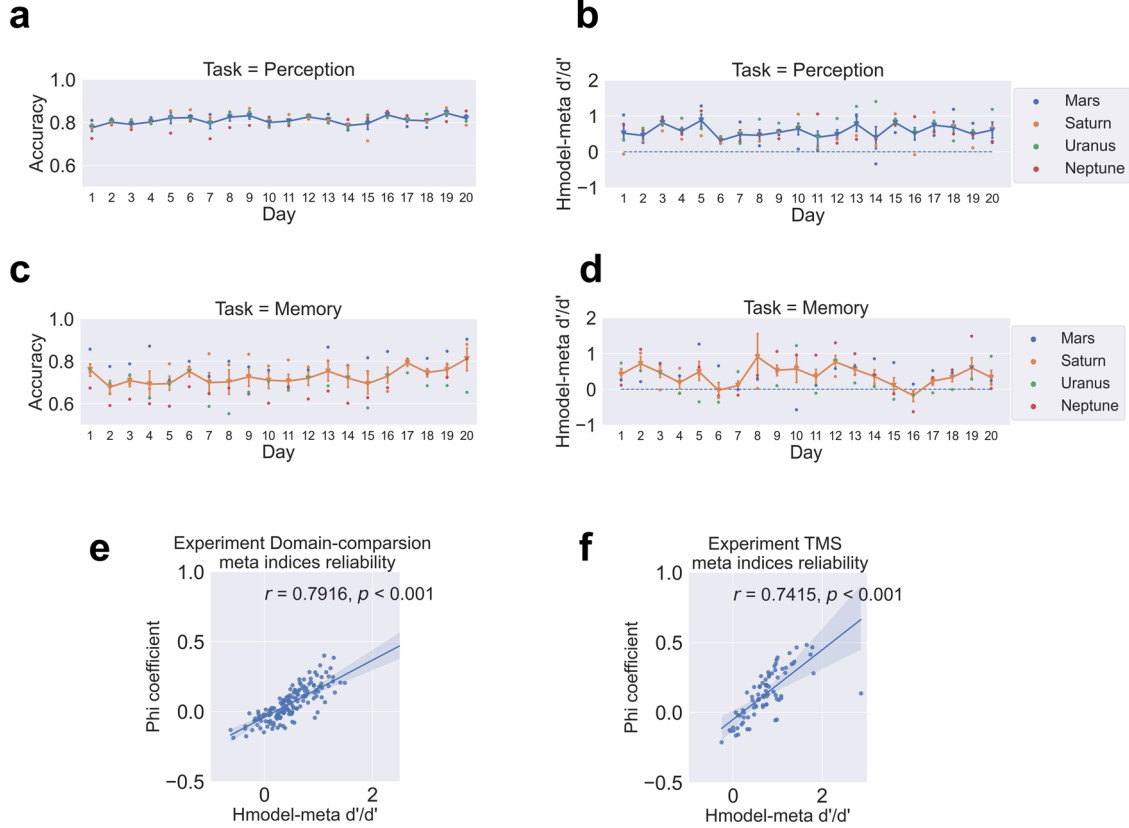

**Fig. 1 Task performance and metacognitive capability remained steady across days.** Plots depict daily accuracy (**a**, **c**) and metacognitive efficiency (**b**, **d**) across 20 days for four monkeys performing two tasks. Strong correlations between the two meta-cognitive metrics (**e**, **f**). Pearson correlations computed among the two meta-indices were statistically significant (both $P$s < 0.001). Error bars indicate ± one standard error.

**Table 1 Percentiles of each monkey's meta-scores compared with the simulated data.**

| Monkey | Memory | | Perception | |
|---|---|---|---|---|
| | Phi | H-model meta d'/d' | Phi | H-model meta d'/d' |
| Mars | 99 | 79 | 99 | 99 |
| Saturn | 97 | 83 | 99 | 98 |
| Uranus | 98 | 86 | 99 | 94 |
| Neptune | 99 | 80 | 99 | 99 |
| Statistics | $0.03^4 < 0.001$ | $0.17^4 < 0.001$ | $0.01^4 < 0.001$ | $0.06^4 < 0.001$ |

Inferential statistics calculated using a minimum statistics method show that the meta-scores of all monkeys are significantly higher than chance level.

**TMS of BA46d impairs metacognitive performance but not cognitive performance**. We then turned to our main question. We tested whether TMS of BA46d would affect metacognition on perceptual decision-making. We performed a 2 (TMS phase: on-judgement/on-wagering) × 2 (TMS: TMS-46d/TMS-sham) mixed-design repeated-measures ANOVA for metacognitive efficiency with TMS phase as a within-subjects factor and TMS as a between-subjects factor. We found a significant interaction between TMS phase and TMS modulation in both monkeys (Neptune, $F_{(1,18)} = 6.431$, $p = 0.021$; Uranus, $F_{(1,18)} = 10.718$, $p = 0.004$). The interaction was driven by lower metacognitive efficiency following TMS of BA46d than following sham treatment in the on-judgement phase condition (paired $t$-tests: Neptune, $t_{(9)} = 3.675$, $p = 0.002$; Uranus, $t_{(9)} = 2.741$, $p = 0.013$), whereas no difference in metacognitive efficiency was found in the on-wagering phase (paired $t$ tests: Neptune, $t_{(9)} = -0.3$, $p = 0.768$; Uranus, $t_{(9)} = -0.841$, $p = 0.411$); see Fig. 2a, b. We replicated the metacognition deficit in the on-judgement phase

with the phi coefficient (paired $t$-tests: Neptune, $t_{(9)} = 3.51$, $p = 0.002$; Uranus, $t_{(9)} = 5.637$, $p < 0.001$).

These meta-indices are based on how the subjects rate their confidence and reflect how meaningful a subject's confidence (reflected here by time wagering) is in distinguishing between correct and incorrect responses. Accordingly, we performed a three-way ANOVA (TMS phase: on-judgement/on-wagering × TMS: TMS-46d/TMS-sham × Confidence: unreached/reached) on task performance (accuracy) and observed a significant three-way interaction in both monkeys (Neptune, $F_{(1,2313)} = 5.530$, $p = 0.019$; Uranus $F_{(1,2295)} = 6.910$, $p = 0.009$). The TMS effect was stronger in the on-judgement TMS phase (TMS × Confidence interaction: Neptune, $F_{(1,1167)} = 10.672$, $p = 0.001$; Uranus $F_{(1,1160)} = 10.404$, $p < 0.001$, Fig. 2c) than in the on-wagering TMS phase (TMS × Confidence interaction): Neptune, $(F_{(1,1146)} = 0.003$, $p = 0.954$; Uranus $F_{(1,1135)} = 0.309$, $p = 0.579$; Fig. 2d). The effects in the on-judgement TMS phase were driven by higher accuracy following TMS-46d than TMS-sham in the unreached trials

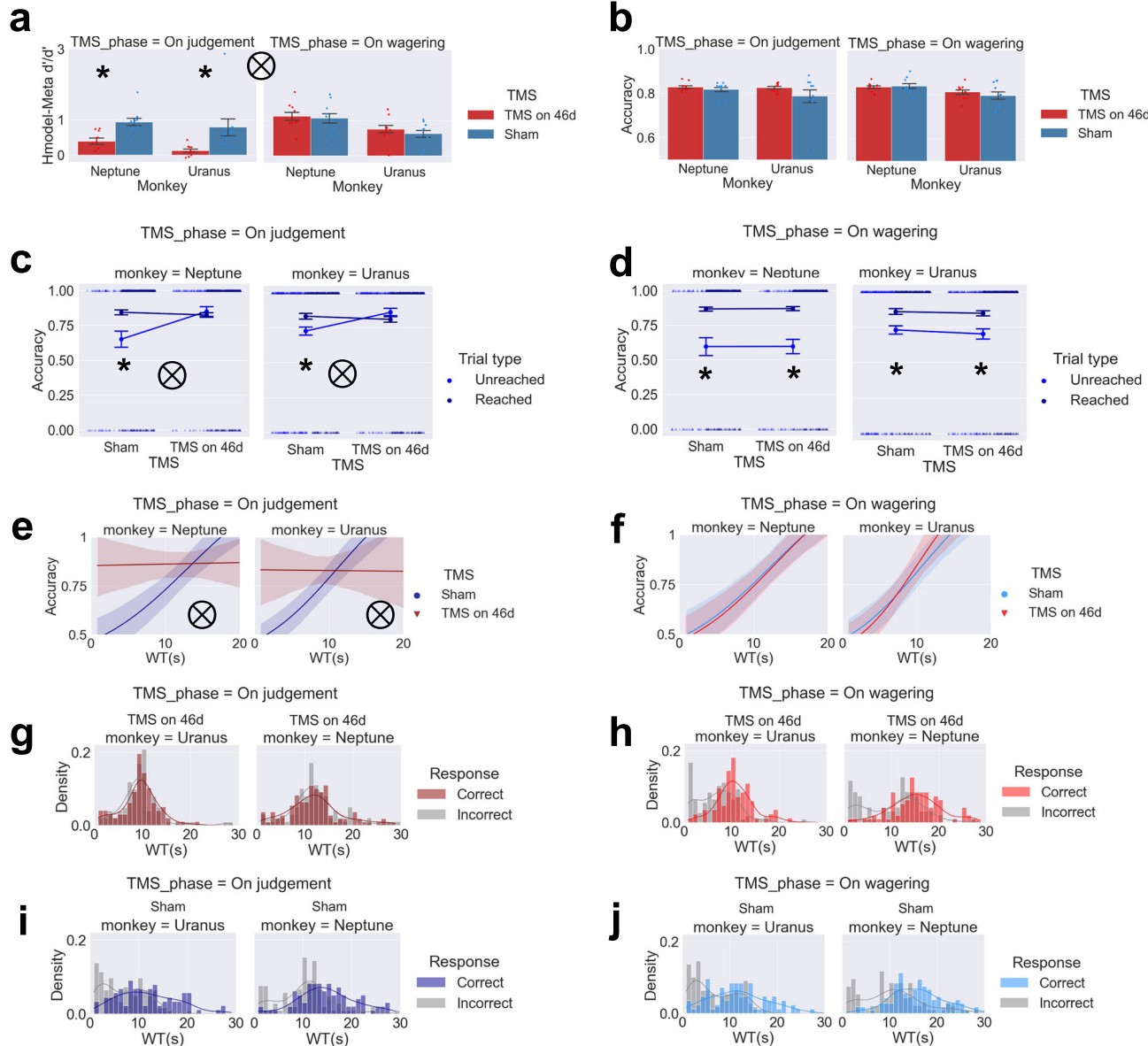

**Fig. 2 TMS during the on-judgement phase disrupts metacognition and the response outcome tracking ability of wagered time (WT).** The monkeys demonstrated an impairment in metacognitive efficiency in the TMS-46d condition during the on-judgement phase but not during the on-wagering phase (**a**). TMS of BA46d does not affect task accuracy (**b**). Difference in accuracy between unreached trials (low confidence) and reached trials (high confidence) in the on-judgement phase and the on-wagering phase (**c**, **d**, respectively). The trendlines are fitted to accuracy by logistic regression with WT as a factor for the TMS-sham and TMS-46d conditions separately. WT reliably tracks response outcomes in the TMS-sham condition but not in the TMS-46d condition during the on-judgement phase. WT tracks response outcomes in both the TMS-sham and TMS-46d conditions during the TMS on-wagering phase (**e**, **f**). Distributional differences between correct and incorrect WT. The largest effects were observed in the TMS-sham condition, in which the BA46d was not perturbed (**g–j**). The WT bin size was set to 1 s; coloured lines indicate kernel density estimation. Error bars indicate ± one standard error; * indicates $p < 0.05$. ⊗ indicates a significant interaction effect ($p < 0.05$) of WT and TMS (TMS-46d/sham). Shaded areas indicate bootstrap-estimated 95% confidence intervals for the regression estimates.

(Mann–Whitney U tests: Neptune, $p = 0.001$; Uranus, $p < 0.001$) but not in the reached trials (Mann-Whitney U tests: Neptune, $p = 0.235$; Uranus, $p = 0.192$). These findings confirmed that TMS targeting BA46d impairs metacognitive ability on a trial-by-trial level.

We further verified that type 1 task performance and mean wagered time were not affected by TMS. As expected, task performance (daily accuracy), reaction time (RT), and wagered time (WT) were not different between the two TMS conditions in either the on-judgement phase (paired $t$ test, all $p$ values >0.1 for accuracy, RT, and WT in both monkeys) or the on-wagering

phase (paired $t$ test, all $p$-values >0.1 for accuracy, RT, and WT in both monkeys).

**Instantiation of TMS-induced impairment: Reduced accuracy-tracking ability of wagered time, altered reaction time–wagered time association, and altered trial-difficulty psychometric curve.** We examined whether TMS would affect the ability of WT to track task performance in the two TMS phases (on-judgement/on-wagering). We focused our analysis on catch trials and incorrect trials, since we could not measure the precise WT for

**Table 2 Individual fitting of data from the TMS experiment by logistic regression.**

| Coefficients | Estimate | Standard Error | Odds Ratio | z | p |
|---|---|---|---|---|---|
| Monkey = Neptune TMS phase = on judgement | | | | | |
| (Intercept) | −1.929 | 0.405 | 0.145 | −4.767 | <0.001 |
| WT | 0.149 | 0.030 | 1.160 | 4.947 | <0.001 |
| TMS | 1.989 | 0.532 | 7.309 | 3.735 | <0.001 |
| WT * TMS | −0.146 | 0.040 | 0.864 | −3.643 | <0.001 |
| Monkey = Uranus TMS phase = on judgement | | | | | |
| (Intercept) | −1.930 | 0.328 | 0.145 | −5.881 | <0.001 |
| WT | 0.174 | 0.031 | 1.190 | 5.541 | <0.001 |
| TMS | 1.905 | 0.492 | 6.719 | 3.875 | <0.001 |
| WT * TMS | −0.175 | 0.048 | 0.83 | −3.670 | <0.001 |
| Monkey = Neptune TMS phase = on wagering | | | | | |
| (Intercept) | −1.816 | 0.400 | 0.163 | −4.539 | <0.001 |
| WT | 0.147 | 0.028 | 1.158 | 5.206 | <0.001 |
| TMS | −0.138 | 0.579 | 0.871 | −0.239 | 0.811 |
| WT*TMS | 0.008 | 0.041 | 1.008 | 0.184 | 0.854 |
| Monkey = Uranus TMS phase = on wagering | | | | | |
| (Intercept) | −1.867 | 0.336 | 0.155 | −5.551 | <0.001 |
| WT | 0.175 | 0.032 | 1.191 | 5.439 | <0.001 |
| TMS | −0.345 | 0.541 | 0.708 | −0.638 | 0.524 |
| WT*TMS | 0.048 | 0.054 | 1.049 | 0.883 | 0.377 |

Logistic regression of response (correct/incorrect) with WT, TMS (TMS-46d/TMS-sham), and a cross-product item as factors to test whether TMS of BA46d affects the ability of WT to track responses. Logistic regression was performed for the on-judgement and on-wagering phases separately for each monkey.

some trials (i.e., correct reached trials; see methods). We performed logistic regression on correctness with WT, TMS (TMS-46d/TMS-sham), and cross-product items as factors to test whether TMS of BA46d affected the response-tracking precision of WT. We found a significant interaction between TMS and WT in the on-judgement TMS phase (both monkeys: $\beta_3 = -0.149$, standard error = 0.029, odds ratio = 0.862, z = −5.115, p < 0.001, Fig. 2e) but not during the on-wagering phase (both monkeys: $\beta_3 = 0.010$, standard error = 0.030, odds ratio = 1.010, z = 0.321, p = 0.748, Fig. 2f). This effect in the on-judgement phase was driven by higher WT in correct trials than in incorrect trials in the TMS-sham condition (Mann–Whitney U tests: Neptune, p < 0.001; Uranus, p < 0.001, Fig. 2i, j) but not in the TMS-46d condition (Mann–Whitney U tests: Neptune, p = 0.98; Uranus, p = 0.45, Fig. 2g). We also confirmed that WT can predict the trial outcomes in a graded manner in the on-wagering phase ($\beta_1 = 0.152$, standard error = 0.020, odds ratio = 1.164, z = 7.631, p < 0.001). These results revealed that TMS of BA46d, when administered during the on-judgement phase, affects metacognitive performance. We obtained the same results when we performed these logistic regressions on the two monkeys separately (Table 2).

Second, metacognitive abilities in animals are often confounded by behavioural association[37]. For example, animals are believed to make use of cues (environmental cues such as stimulus conditions and self-generated cues such as response time) to determine confidence instead of performing the task metacognitively. To rule out this possibility, we calculated the correlation between RT and WT in both experiments to check whether the monkeys relied on RT as an associative cue to determine confidence. The results showed no correlation between RT and WT correlation in the domain-comparison experiment (Fig. 3a), indicating that the macaques did not rely on RT as an associative cue to determine their WT. We then utilized this phenomenon to verify the effect of TMS. WT was significantly negatively correlated with RT during the on-judgement TMS phase only in the TMS-46d condition (r = −0.195, p < 0.001) and not in the TMS-sham condition (Fig. 3b). We found a significant difference in correlation coefficients between TMS-46d and TMS-sham in the on-judgement phase (z = −2.24, p = 0.0251). It is

possible that monkeys started to rely on RT as an associative cue after having received TMS on BA46d, which hampered their metacognitive ability. As a control comparison, no difference was found between TMS conditions in the on-wagering phase (Fig. 3c). Moreover, we found a strong negative correlation (point biserial correlation) between accuracy and RT. Specifically, we showed RT was negatively correlated with accuracy in the domain-comparison experiment (perception, r = −0.0819, p < 0.001; memory, r = −0.17535, p < 0.001; Fig. 3d), and in both on-judgement (TMS-46d, r = −0.0856 p = 0.0038; Sham, r = −0.1345, p < 0.001; Fig. 3e) and on-wagering (TMS-46d, r = −0.0983, p < 0.001; Sham, r = −0.1063, p < 0.001; Fig. 3f) phase in the TMS experiment. We also found a negative correlation in correct trials (r = −0.266, p < 0.001; Fig. 3g), and a negative correlation tendency in incorrect trials (r = −0.1064, p = 0.1336; Fig. 3h) in the TMS-46d condition.

Moreover, as seen in the rodent literature, WT can be expressed as a function of the strength of evidence (e.g., odour mixture ratio in their task) and response outcome (correct/incorrect);[20] the level of confidence should increase with evidence strength (resolution difference in our experiments) for correct trials and decrease with evidence strength for incorrect trials. We performed GLM to predict WT with four variables: TMS (TMS-46d/TMS-sham), TMS phase (on-judgement/on-wagering phase), resolution difference, and correctness and their cross-product items. We found a four-way interaction in the monkeys (Neptune, $\beta_{\text{TMS} \times \text{TMS phase} \times \text{correctness} \times \text{resolution difference}} = -60.66$, p = 0.010; Uranus, $\beta_{\text{TMS} \times \text{TMS phase} \times \text{correctness} \times \text{resolution difference}} = -44.76$, p = 0.019). Trial-difficulty psychometric curves of these results illustrated that the effects were driven by a strengthened correctness × resolution difference interaction in the TMS-sham condition (including trials in both the on-judgement TMS phase and the on-wagering TMS phase) (Neptune, $\beta_{\text{correctness} \times \text{resolution difference}} = 48.99$, p < 0.001; Uranus, $\beta_{\text{correctness} \times \text{resolution difference}} = 42.20$, p < 0.001) and no effect in the TMS-46d on-judgement condition (Neptune, $\beta_{\text{correctness} \times \text{resolution difference}} = 13.55$, p = 0.119; Uranus, $\beta_{\text{correctness} \times \text{resolution difference}} = -2.50$, p = 0.753, Fig. 4c). We also found a Pearson correlation between WT and task difficulty in the TMS experiment (for two monkeys: r = −0.062, p = 0.010; Uranus, r = −0.0710, p = 0.046; Neptune, r = −0.108, p = 0.002).

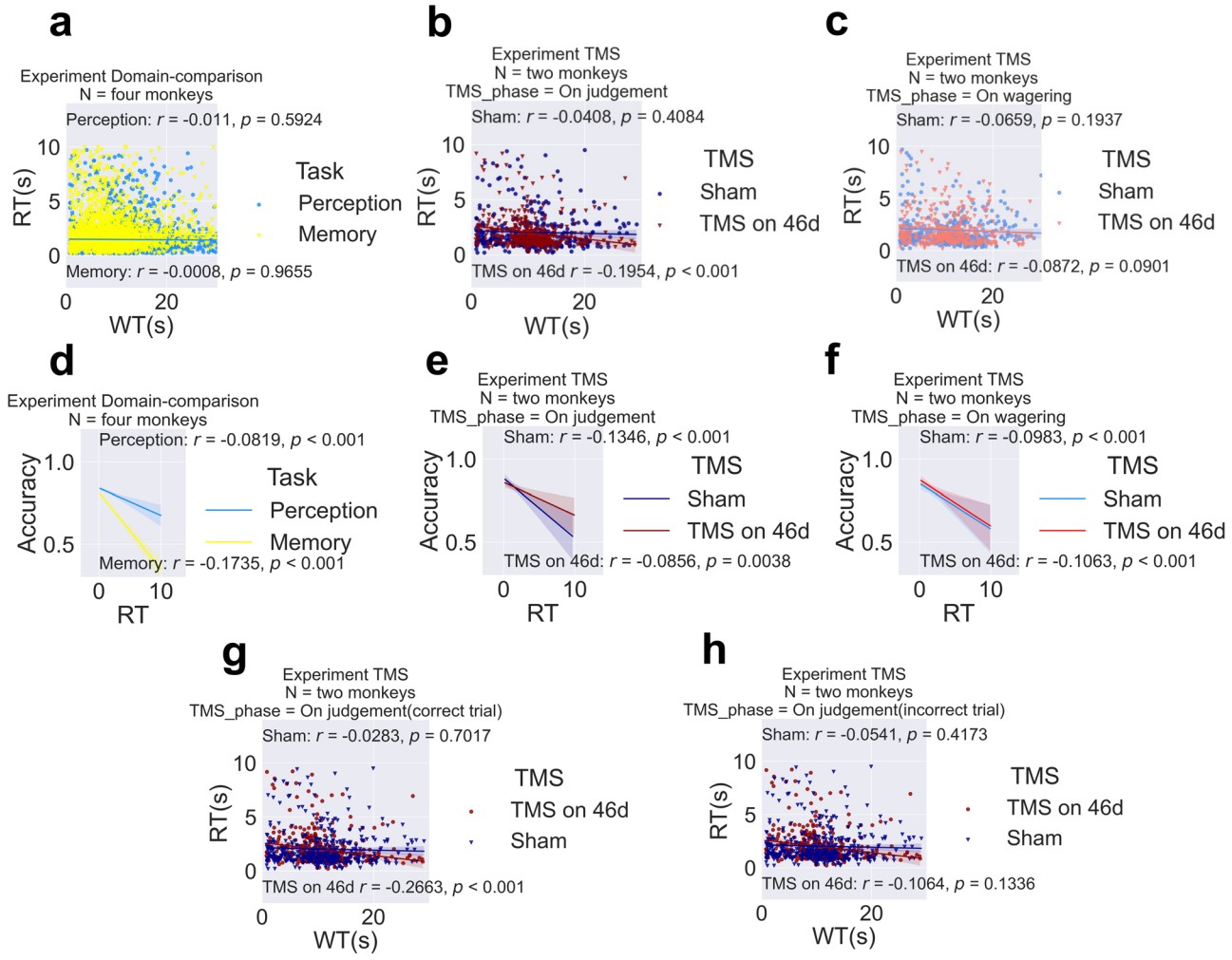

**Fig. 3 On-judgement TMS alters the correlation between reaction time (RT) and wagered time (WT).** No correlation was found between RT and WT in the domain-comparison experiment (**a**). The Pearson correlation between RT and WT during the on-judgement phase was statistically significant for the TMS-46d condition ($p < 0.001$) but not significant for the TMS-sham condition (**b**). The correlations during the on-wagering phase were not significant for either TMS condition (**c**). RT was significantly negatively correlated with accuracy (correct/incorrect) in the domain-comparison experiment (**d**) and in both TMS phases in the TMS experiment (**e**, **f**). A negative correlation between RT and WT in TMS-46d condition in correct trials (**g**) but not in incorrect trials (**h**). Shaded areas indicate bootstrap-estimated 95% confidence intervals for the regression estimates.

Critically, the correctness × resolution difference interaction was driven by the increased WT for correct trials in the TMS-sham condition (including trials in both the on-judgement TMS phase and the on-wagering TMS phase) (Neptune, $\beta_{resolution\ difference} = 27.47$, $p < 0.001$; Uranus, $\beta_{resolution\ difference} = 27.76$, $p < 0.001$) and decreased WT for incorrect trials (Neptune, $\beta_{resolution\ difference} = -21.51$, $p < 0.001$; Uranus, $\beta_{resolution\ difference} = -14.43$, $p < 0.001$, Fig. 4d–f). These results suggest that in the TMS-sham condition, WT increased with resolution difference for correct trials and decreased with resolution difference for incorrect trials irrespective of TMS phase, whereas this pattern was disrupted during the on-judgement phase in the TMS-46d condition. Additionally, we confirmed that perceptual performance was intact by performing logistic regression on response outcomes with resolution difference, TMS (TMS-46d/TMS-sham), and cross-product item as factors. We found no interactions for either the on-judgement TMS phase or the on-wagering TMS phase in the monkeys (all $Ps > 0.05$). As the performance accuracy was controlled by a staircase procedure, we compared the distributional differences between the TMS conditions and we did not find significant differences in task difficulty (resolution difference here) between TMS-46d and TMS-sham conditions in either on-judgement phase (Mann–Whitney U test

results: Uranus, $p = 0.074$; Neptune, $p = 0.804$; Fig. 4g) or on-wagering phase (Mann–Whitney U test results: Uranus, $p = 0.158$; Neptune, $p = 0.635$; Fig. 4h). In terms of accuracy, we managed to keep the overall performance accuracy in the range of 62.6–86.3% (mean: 81.7% ± 3.6%), which is within a reasonable range compared to a recommended accuracy (cf. ~71% as discussed in two stuides[1,38]). We also believe that if the monkeys reached ceiling in accuracy, the metacognitive judgement shall be skewed, and the chance leading to differences between TMS-sham and TMS-46d conditions would be very negligible. These findings confirmed our first hypothesis that the monkey dlPFC is critical for meta-perception and that such effects are independent of perception task processes.

**Qualities of monkey metacognition: Wagered time (WT) is diagnostic of the animals' performance.** To further substantiate these results, we expected that monkeys could indicate their confidence using their trial-by-trial wagered time. We showed that wagered time is diagnostic of the animals' performance using a number of analyses. First, we compared the accuracy in reached (high confidence) and unreached (low confidence) trials; chi-square tests revealed that monkeys had higher accuracy in

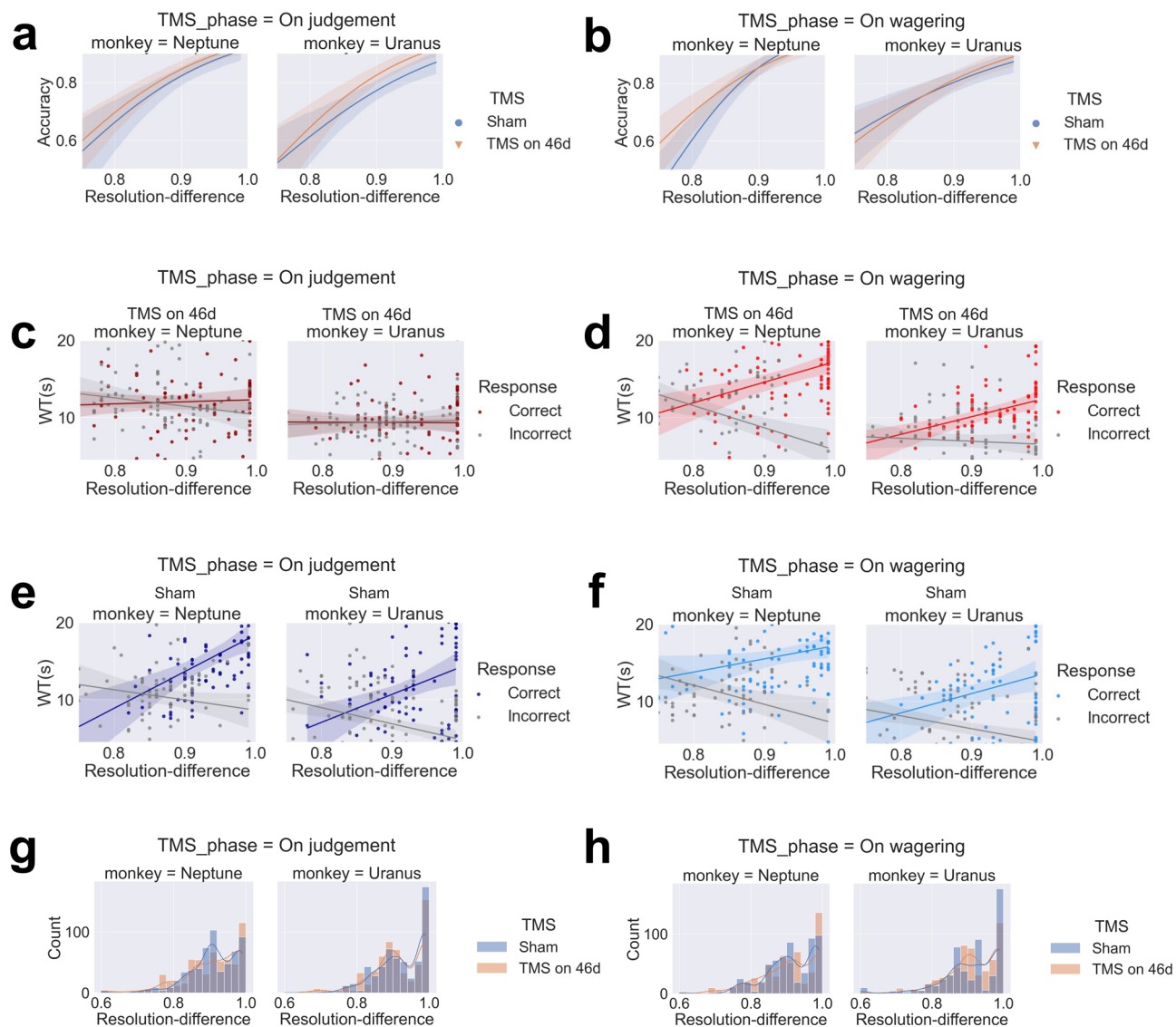

**Fig. 4 On-judgement TMS distorts the trial-difficulty psychometric curve.** Accuracy decreases with task difficulty (resolution difference; higher values indicate lower task difficulty). The lines are logistic regression fits for accuracy with resolution difference as a factor, calculated separately for the TMS-sham and TMS-46d conditions in the on-judgement phase (**a**) and on-wagering phase (**b**). WT decreased with task difficulty in correct trials and increased with task difficulty in incorrect trials in all control conditions (**d**, **f**), but this pattern was absent in the on-judgement phase of the TMS-46d condition (**c**). Distributional differences between TMS-46d and TMS-sham conditions were not significant in either on-judgement phase (**g**) or on-wagering phase (**h**), indicating task difficulty were well controlled. Resolution-difference bin size set to 0.02; colored lines indicate kernel density estimation. Shaded areas indicate bootstrap-estimated 95% confidence intervals for the regression estimates.

higher-confidence trials in both meta-perception (all four monkeys: $\chi^2_{(1)} = 31.88$, $p < 0.001$; for individual monkeys: all $p$ values $< 0.05$, Fig. 5a) and meta-memory (all four monkeys: $\chi^2_{(1)} = 13.41$, $p < 0.001$; for individual monkeys: all $p$ values $< 0.05$, Fig. 5b). To test whether the WT tracked the response outcomes, we performed logistic regression on response outcomes with WT, task (memory/perception), and the cross-product as factors. We confirmed that the WT could accurately predict the trial outcome ($\beta_1 = 0.033$, standard error $= 0.007$, odds ratio $= 1.033$, $z = 4.586$, $p < 0.001$; Fig. 5e). We found no interaction between task and WT ($\beta_3 = 0.0014$, standard error $= 0.011$, odds ratio $= 1.014$, $z = 1.335$, $p = 0.182$), indicating that WT in both memory and perception tasks tracked the response outcomes. These results showed that the trial-wise wagered time was diagnostic of the animals' decision outcome, reflecting that the monkeys were aware of their judgement outcome. All results held when we performed the analyses for each monkey individually (Table 3).

**Qualities of monkey metacognition: Evidence regarding domain specificity.** While we found a positive correlation between the perception and memory domains in daily individual accuracy ($r_{(80)} = 0.271$; $p = 0.0151$; Fig. 6a), their respective metacognitive efficiency scores did not correlate ($r_{(80)} = 0.1134$; $p = 0.3164$; right panel in Fig. 6b). This prompted us to examine the domain specificity with bias-free metacognitive efficiency (H-model $meta\text{-}d'/d'$). To assess the potential covariation between metacognitive abilities, we calculated a domain-generality index (DGI) for each subject. We quantified each monkey's domain generality as well as the mean across the two tasks (Fig. 6c, d). Specifically, we shuffled the task types (memory/perception) across all 40 days (20 days of memory and 20 days of perception) within each subject. This procedure was shuffled 1000 times, and we obtained 40,000 simulated meta-index values for each monkey. We found that all monkeys' DGIs were above the simulated values, as confirmed by Mann–Whitney U tests against the

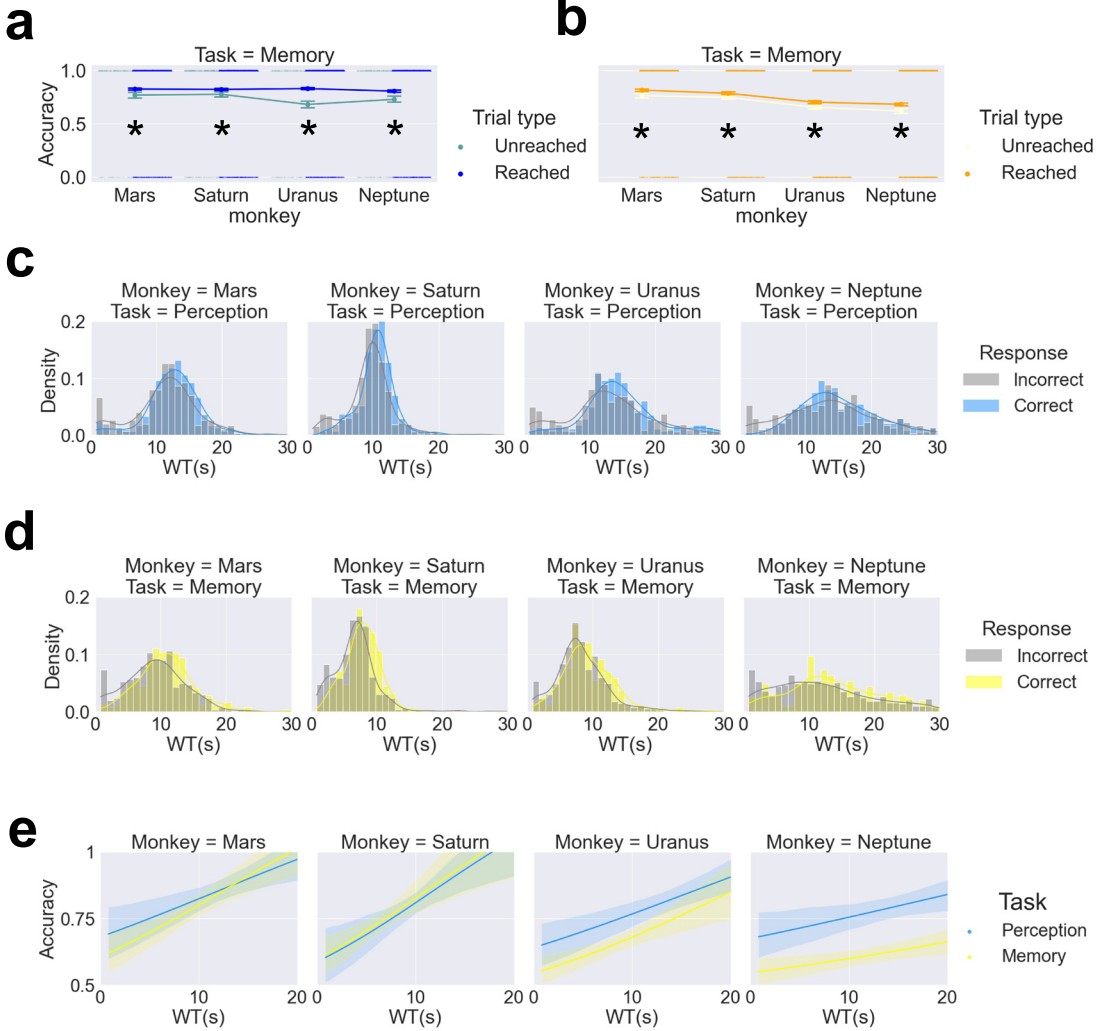

**Fig. 5 Wagered time reflects monkeys' task performance (correctness) in both memory and perception tasks.** Difference in accuracy between unreached trials and reached trials in the perception (**a**) and memory tasks (**b**). Differences between the WTs of correct and incorrect trials for each monkey in the perception (**c**) and memory tasks (**d**). WT tracks response outcome (correct/incorrect) in both memory and perception tasks. The lines are logistic regression fits for accuracy with WT as a factor. The WT bin size was set to 1 s; coloured lines indicate kernel density estimation (**e**). Error bars indicate ± one standard error; * indicates $p < 0.05$. Shaded areas indicate bootstrap-estimated 95% confidence intervals for the regression estimates.

simulated data (the mean of simulated data, Mars: 0.167; Saturn: 0.182; Uranus: 0.350; Neptune: 0.260; Mann–Whitney U test results: all $p$ values < 0.001, Fig. 6e). In order to be compatible with the literature[4,6], we used the absolute values for this analysis. However, we also confirmed the results with signed DGI. We performed Mann–Whitney U test on signed DGI and replicated the domain-specific effects (all four monkeys, $p < 0.001$; Mars, $p = 0.153$; Saturn, $p < 0.001$; Uranus, $p < 0.001$; Neptune, $p = 0.263$). Additionally, we employed pairwise correlation to assess the similarity of the two tasks across and within subjects separately for Hmodel-*meta d'/d'* (Fig. 6g) and accuracy (Fig. 6i). The matrix of pairwise correlation was hierarchically clustered for Hmodel-*meta d'/d'* (Fig. 6h), revealing two distinct clusters in which data from the same domain in multiple monkeys grouped together (whereas within-monkey data did not). We calculated the standardized Euclidean distance of each vector pair for Hmodel-*meta d'/d'* and accuracy (in total 28 vector pairs, each vector corresponding to each row in Fig. 6g–j, each row containing 8 cells) and found pairwise distance of Hmodel-*meta d'/d'* in within-tasks across monkeys are significantly shorter than across-tasks within monkeys (Mann–Whitney U test results: $p = 0.033$), but not for pairwise distance of accuracy

(Mann–Whitney U test results: $p = 0.380$). This indicates that the within-task similarity of metacognitive efficiency was stronger than the within-subjects similarity. Together, these results suggest domain-specific constraints on metacognitive ability that transcend the individual animal level.

## Discussion

Our findings on deficits following TMS of BA46d demonstrate functional and biological dissociation of cognition and metacognition in animals[16,18]. Together with evidence of metacognitive domain specificity, our results characterize the specialization of metacognition in primates.

The TMS-induced metacognitive deficit revealed here is specific to the correspondence between accuracy and confidence (cf. criteria for producing subjective ratings[10]) rather than to the animals' task performance (RT or accuracy). Mechanistically, TMS affects neural functioning by inducing a short-lasting electric field at suprathreshold intensities via electromagnetic induction[39]. By combining T1-weighted imaging with a stereotaxic system, we reliably confined the focus of the stimulation to BA46d (with some stimulation possibly reaching subregions in the dlPFC, e.g., 9m, 9d, 46v, and 46f). Our results corroborate the

**Table 3 Individual fitting of data from the domain-comparison experiment by logistic regression.**

| Coefficients | Estimate | Standard Error | Odds Ratio | z | p |
|---|---|---|---|---|---|
| Monkey = Mars | | | | | |
| (Intercept) | −1.082 | 0.225 | 0.339 | −4.800 | <0.001 |
| Task | 0.365 | 0.328 | 1.440 | 1.112 | 0.266 |
| WT | 0.087 | 0.021 | 1.091 | 4.207 | <0.001 |
| Task * WT | −0.027 | 0.028 | 0.973 | −0.974 | 0.330 |
| Monkey = Saturn | | | | | |
| (Intercept) | −1.127 | 0.196 | 0.324 | −5.746 | <0.001 |
| Task | −0.053 | 0.347 | 0.948 | −0.153 | 0.879 |
| WT | 0.102 | 0.025 | 1.107 | 4.081 | <0.001 |
| Task * WT | −0.002 | 0.037 | 0.998 | −0.046 | 0.964 |
| Monkey = Uranus | | | | | |
| (Intercept) | −1.435 | 0.178 | 0.238 | −0.8060 | <0.001 |
| Task | 0.530 | 0.279 | 1.699 | 1.898 | 0.058 |
| WT | 0.071 | 0.018 | 1.074 | 3.914 | <0.001 |
| Task * WT | −0.016 | 0.023 | 0.985 | −0.668 | 0.504 |
| Monkey = Neptune | | | | | |
| (Intercept) | −1.428 | 0.166 | 0.240 | −8.596 | <0.001 |
| Task | 0.685 | 0.274 | 1.984 | 2.499 | 0.012 |
| WT | 0.031 | 0.011 | 1.032 | 2.825 | 0.005 |
| Task*WT | 0.003 | 0.018 | 1.003 | 0.187 | 0.851 |

Logistic regression of response (correct/incorrect) with WT, task (memory/perception), and a cross-product item as factors to test whether WT tracks responses. The results show that the response outcomes were tracked by WT. Logistic regression was performed separately for each monkey.

human literature. The human lateral PFC has been associated with a unique type of metacognitive process—the feeling of knowing[14]. Studies inactivating the dlPFC to diminish metacognitive ability without altering perceptual discrimination performance and confidence criteria[10], as well as decoded multivariate patterns in the lPFC pertaining to metacognitive judgements, indicate the lPFC's involvement in conscious experiences[6]. Our results confirmed that the dorsal part of the lPFC in monkeys plays a critical role in mediating perceptual experiences. We should note that the metacognitive functions of the lPFC are distinct from the neuronal activity in the LIP[21], supplementary eye field (SEF)[15], and middle temporal visual area (MT)[40], which have been shown to carry information that correlates with both perceptual decisions and metacognition. Our results are in line with the view that the general role of the dlPFC lies in information monitoring and maintenance[38,41]. It is possible that the neural signal changes status from first-order representations to higher-order representations[8], which enables the perceptual content to enter consciousness.

In terms of the temporal window of meta-computation, by applying high-temporal-resolution TMS to the monkey dlPFC in the on-judgement and on-wagering phases, we revealed that meta-calculation processes were carried out in the relatively early stage. This is in line with findings that the LIP in monkeys computes perceptual evidence at the time of judgement[22]. However, interestingly, the human aPFC[42] and dorsal premotor cortex[22] along with the rodent OFC[18,20] support late-stage meta-calculation. For example, single neurons in the OFC of rodents showed neural activity that predicted the trial-difficulty psychometric curve during wagering[20], indicating the role of the OFC in late-stage meta-calculation. Some computational models have also proposed that post-decisional (late-stage) processes are essential for meta-calculation[43,44]. To tap further into these issues, a recent study applied online TMS pulses (three consecutive pulses: 250, 350, and 450 ms after stimulus onset) to the human dlPFC and showed that TMS alters subjective confidence but not metacognitive ability[12]. By comparing their TMS timing with ours, it can be inferred that processes necessary for meta-calculation might have happened earlier than those required for confidence calculation (TMS at 250 ms led to deficits in confidence calculation, whereas TMS at

100 ms led to deficits in meta-calculation in our study). In this case, the dlPFC performs meta-calculation at approximately 100–250 ms and permits the confidence expression at a later stage. The very short duration (100–250 ms) during which meta-calculation could be affected seems to suggest that meta-calculation is heuristic[45]. However, we hold the opinion that the precise timing for perceptual decision-making is not entirely clear. Here, we made the assumption that meta-calculation could be an integral part of first-order decision-making. We, therefore, set the TMS timing as close to the stimuli onset as possible (that is, 100 ms after stimuli-onset). Be transient the TMS evoked potential (TEP) as it may, we obtained a strong dissociation between meta-indices and accuracy, hinting a possibility that the dlPFC performs meta-calculation at a very early stage. In contrast to humans, whose metacognitive ability can be assessed by quantifying trial-by-trial correspondence between objective performance and subjective confidence[46–49], studies on animals have used binary means of confidence expression such as betting[15–17,29,30,45,50], opt-out[21,29,51–53], or some secondary metrics such as reaction times[54,55] and saccadic endpoints[52]. However, binary reports have several shortcomings. For example, we cannot preclude the possibility that information is integrated before reporting, merging various putative processes underlying metacognitive control[56] and monitoring[57,58]. Since the relationship between response and confidence is affected by distribution assessments[59], binary or even scaled confidence reports will make it impossible to obtain a confidence distribution[24]. As a result, information falling within the intermediate confidence range in the calibration of confidence and accuracy will also be missed[60–62]. For these considerations, we, therefore, adopted Lak et al.'s[18] paradigm and provided a quantitative and continuous proxy for confidence akin to self-reporting in humans.

The results obtained with this paradigm allowed us to address a long-standing controversy in the animal cognition literature. Previous studies have established that several other species are capable of monitoring their own behaviour[21,29,51,53,63–67]. However, due to the extensive training that is often required, animals' metacognitive ability can be confounded by various types of cue associations[37]. Importantly, with the temporal wagering

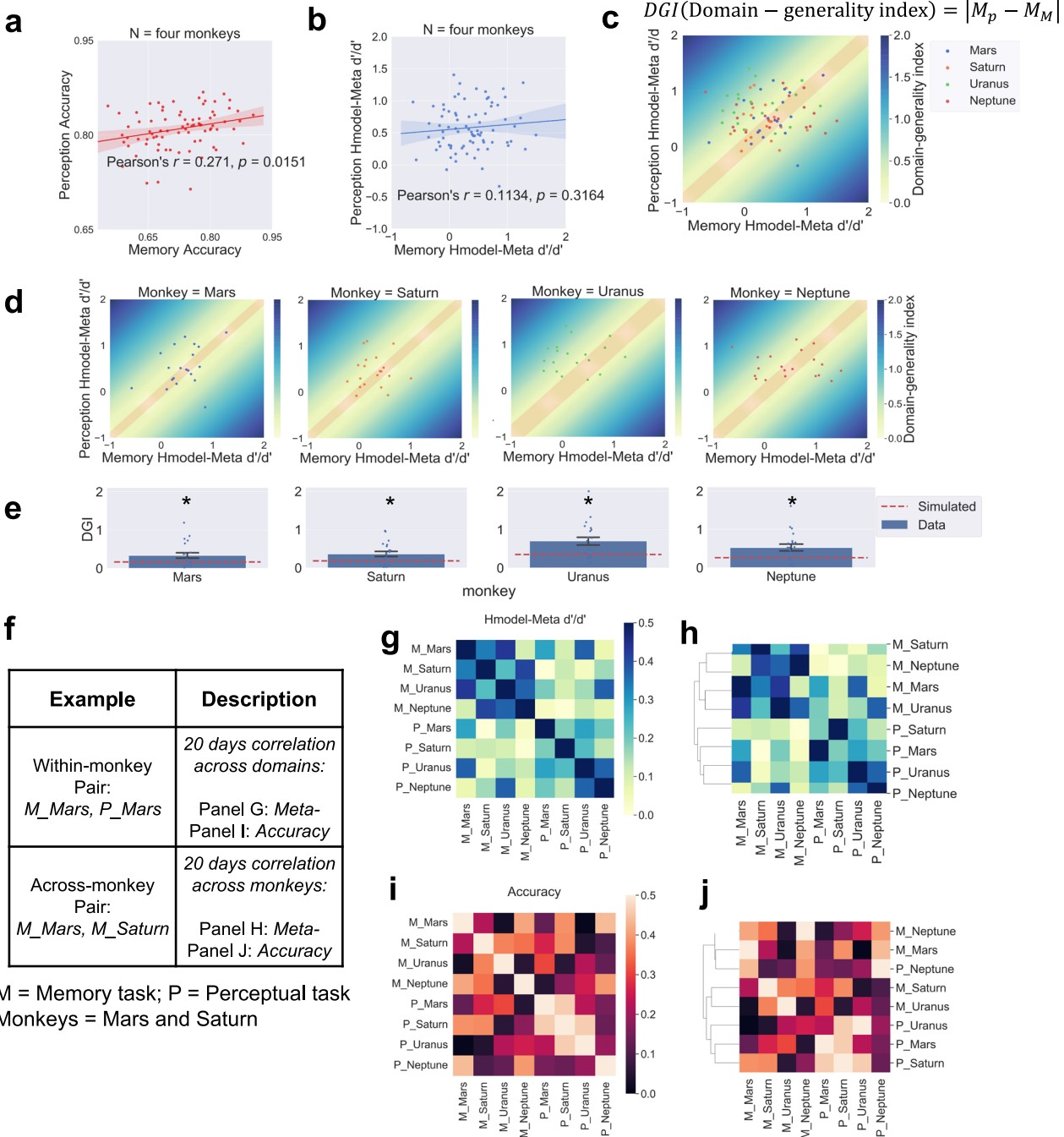

**Fig. 6 Domain-specific metacognition in monkeys.** Task performance in terms of percentage correct was correlated across perceptual and memory domains (**a**). In contrast, their metacognitive efficiency was not correlated across perceptual and memory domains (**b**). The DGI quantifies the similarity between their metacognitive efficiency scores in each domain. Greater DGI scores indicate less metacognitive consistency across domains. Darker colours indicate lower metacognitive generality across domains, and the red area indicates the simulated DGI values. The daily domain-generality index (DGI) is shown for each monkey (**c**) and for all four monkeys (**d**). The monkeys demonstrate a greater DGI than shuffled data (chance) (**e**). Two example pairs for pairwise correlation analysis are described (**f**). The pairwise correlation matrix indicate a pairwise correlation between each monkey and each domain in Hmodel-*meta d′/d′* (**g**) and in accuracy (**i**). Cluster results from the pairwise correlation matrix in Hmodel-*meta d′/d′*, revealing two distinct clusters in which data from the same domain grouped together (**h**), but not in accuracy (**j**). Error bars indicate ± one standard error; * indicates $p < 0.05$. Shaded areas indicate bootstrap-estimated 95% confidence intervals for the regression estimates.

paradigm, the monkeys' introspective knowledge of their memory/perception state in our studies is unlikely to be confounded by these associative factors. The observation that their RT is not associated with WT under normal circumstances shows that monkeys did not use RT as a behavioural cue for wagering decisions[16,65]. Only when BA46d was perturbed did the monkeys

rely on trialwise RT as an associative cue to determine confidence, potentially as a means to compensate for their metacognitive deficits to some extent (note that their metacognitive scores remained above zero in all conditions). This pattern shift suggests that the monkeys might have changed their strategy to rely on external information (e.g., behavioural cues such as RT) when

their introspective ability was suppressed[37], satisfying the established criterion required for animal metacognition.

Our domain-generality index and intraday correlation analysis serve to reveal the existence of such domain-specific metacognition in monkeys. The pairwise correlation shows that the domain specificity is more robust than the within-individual correlation. Behavioural studies have found that efficient metacognition in one task predicts good metacognition in another task[23–28]. Recent studies showed that BOLD signals around BA46d in dlPFC in the macaque brain are associated with metamemory[16,17], whereas our current results showed a causal role of BA46d in dlPFC in meta-perception. This led us to suggest that BA46d might have a domain-general role in metacognition. We also found that the TMS pulse affected on-judgement phase but not on-wagering phase, indicating BA46d is especially functionally related to processes underlying evidence accumulation during decision. It is likely that dlPFC and other regions such as the supplementary eye field (SEF) play different roles at different metacognitive stages. It is known that the supplementary eye field neurons encode metacognitive components in a meta-perception study[15,68]. Moreover, the co-existence of domain-general and domain-specific BOLD signals has been reported in humans[6]. To put this evidence into perspective, we are inclined to postulate that BA46d accumulates domain-general evidence and relates the information to some downstream domain-specific areas such as the SEF. However, indeed, in the current study, it remains possible that we could not precisely demarcate the respective loci for domain-specific metacognition for perception vs. memory since our TMS might have affected a relatively large portion of the dlPFC. Here, we found that monkeys successfully generalized their metacognitive ability from memory to perception (or vice versa). Such generalization suggests that monkeys are capable of using domain-general cues to monitor the status of cognitive processes and assess knowledge states[29,30], carrying theoretical implications for how metacognition and decision confidence are formed in animals.

In summary, we provided evidence for a high-level cognitive faculty in a nonhuman primate species. We pinpointed the critical functional role of BA46d in supporting metacognition independent of task performance, and we found that metacognition in macaques is highly domain-specific for memory versus perception processes.

## Methods
### Experimental protocol
*Animals.* Four male adult macaque monkeys (*Macaca mulatta*, mean age: 6 y; mean weight: 8.2 ± 0.4 kg) took part in this study. They were initially housed in a group of 4 in a spacious, specially designed enclosure (maximum capacity = 12–16 adults) with enrichment elements (e.g., swings and climbing structures). During the experiment, the monkeys were kept in pairs according to their social hierarchy and temperament. They were given individual rations of 180 g monkey chow and pieces of fruit twice a day (9:00 am/3:00 pm). Except on experimental days, the monkeys had unlimited access to water and were routinely given treats such as peanuts and raisins. The monkeys were procured from a nationally accredited colony in the outskirts of Beijing, where the monkeys were bred and reared. The room in which they were housed was illuminated on a 12/12-hour light-dark cycle and was kept at a temperature of 18–23 °C with a humidity of 60–80%. The experimental protocol was approved by the Institutional Animal Care and Use Committee (permission code: M020150902 & M020150902-2018) at East China Normal University.

### Behavioural tasks
*Perception task.* We used resolution difference judgement as our perceptual task[33]; see Fig. 7b. The monkeys began a perceptual trial by touching a blue rectangle in the centre of the screen (which served as a self-paced start cue), and after a variable delay duration (1–6 s), two pictures (which differed in resolution and were shrunken in both length and width) were displayed on opposite sides of the screen. The monkeys were trained to choose and hold onto the target picture (either higher or lower resolution; counterbalanced across monkeys). To maintain stable

cognitive performance across days, we controlled cognitive performance using a 4 up–1 down staircase procedure with resolution difference as a variable.

*Memory task.* We used temporal order judgement as our mnemonic task[69]. Monkeys initiated each memory trial by touching a red rectangle in the centre of the screen, and following a 4-s video clip and a variable delay duration (1–6 s), two frames extracted from the clip were displayed on opposite sides of the screen. Monkeys were trained to choose and hold onto the frame that was shown earlier in the clip. The memory and perception tasks drew from the same pool of pictures, which enabled us to avoid interference from stimulus context, allowing a matched comparison of the memory and perception tasks. The monkeys had been trained extensively for over 6 months on a variant of this memory task, they were already exceedingly stable in their memory performances across the period of testing.

*TMS experimental design (perceptual test only), time schedule, and preliminary training.* Uranus and Neptune received 20 days of meta-perception testing with single-pulse TMS intervention (Uranus: 2303 trials, Neptune: 2321 trials). There were two experimental factors. The first factor was TMS stimulation condition: either TMS was administered to the right BA46d, or sham TMS was performed at the same anatomical site. The second factor was the timing of TMS: in the on-judgement condition, the monkeys received a single pulse 100 ms after stimulus onset, whereas in the on-wagering condition, the monkeys received a single pulse 100 ms after they made their decision (see Fig. 7b). The timing conditions were completed in two within-session blocks (on-judgement, on-wagering) with an interval of 5 minutes between them. The order of TMS-46d/sham and on-judgement/on-wagering was counterbalanced within and across monkeys (Fig. 7a). The TMS experiment was conducted 10 months after the domain-comparison experiment. Two of the monkeys were implanted with head-posts, which was a prerequisite for maintaining their heads steadily for the TMS stimulation, so the TMS experiment data was acquired only from these two monkeys (Uranus and Neptune).

*Domain-comparison experiment: design, time schedule, and preliminary training.* The monkeys were tested for 20 days in the meta-memory task (Saturn: 2165 trials; Neptune: 2196 trials; Mars: 1694 trials; Uranus: 2200 trials) and 20 days for the meta-perception task (Saturn: 1923 trials; Neptune: 2061 trials; Mars: 1851 trials; Uranus: 2087 trials). The testing order for the two tasks was counterbalanced across monkeys: Saturn and Neptune performed the meta-memory task followed by the meta-perception task, whereas Mars and Uranus performed the tasks in the opposite order. Each daily session required the animals to complete 120 trials. All monkeys completed the testing in the allotted time except for Mars, who did not complete enough trials of the meta-memory task on some days. Accordingly, we conducted an extra 10 days of testing on Mars to obtain the number of trials required.

*TMS protocol.* Single-pulse TMS (monophasic pulses, 100 μs rise time, 1 ms duration) was applied using a Magventure X100 (Magventure, Denmark) and an MC-B35 butterfly coil with 35-mm circular components. Based on feasibility analysis of cross-species TMS comparison[70,71], we made use of smaller coils to induce more focal electromagnetic fields to compensate for the small head size of monkeys relative to humans[72]. The pulse intensity was at 120% of the resting motor threshold, which was defined as the lowest TMS intensity that would elicit visible twitches in at least 5 of 10 consecutive pulses when delivered over the right motor cortex[73]. For the stability of the TMS setup, a headpost (Crist Instruments) was affixed to the monkey's skull with screws made of nonmagnetic material. The TMS coil was held in place by an adjustable metal arm. For control purposes, we opted for a sham-condition approach. By this, we rotated the coil 90 degrees over BA46d, thereby ensuring that the sound and vibration (by-products) of the stimulation were identical between the TMS-46d and TMS-sham conditions. Since we have head-posts implanted near the mid-line on the two monkeys, options for control sites (e.g., homologue for the human vertex) were very limited operationally.

*Stimulation sites and localization procedure.* Structural T1-weighted images from post-training MRI scanning were used to enable subject-specific neuronavigation. Brainsight 2.0, a computerized frameless stereotaxic system (Rogue Research), was used to localize the target brain regions. To determine the area of BA46d in each monkey, we first performed nonlinear registration of the T1W images to the D99 atlas and resampled the D99 macaque atlas in native space[74]. Then, the same atlas was used to define each monkey's BA46d. We uploaded each monkey's BA46d mask into the system along with the T1-weighted images for navigation. The stimulated site was located in BA46d (coordinates in monkey atlas: $x = 13$, $y = 16$, $z = 12$) for each monkey (Fig. 7d). To align each monkey's head with the MRI scans, information on the location of each monkey's head was obtained individually by touching three fiducial points, namely, the nasion and the intertragal notch of each ear, using an infrared pointer. The real-time locations of reflective markers attached to the coil and the subject were monitored by an infrared camera with a Polaris Optical Tracking System (Northern Digital).

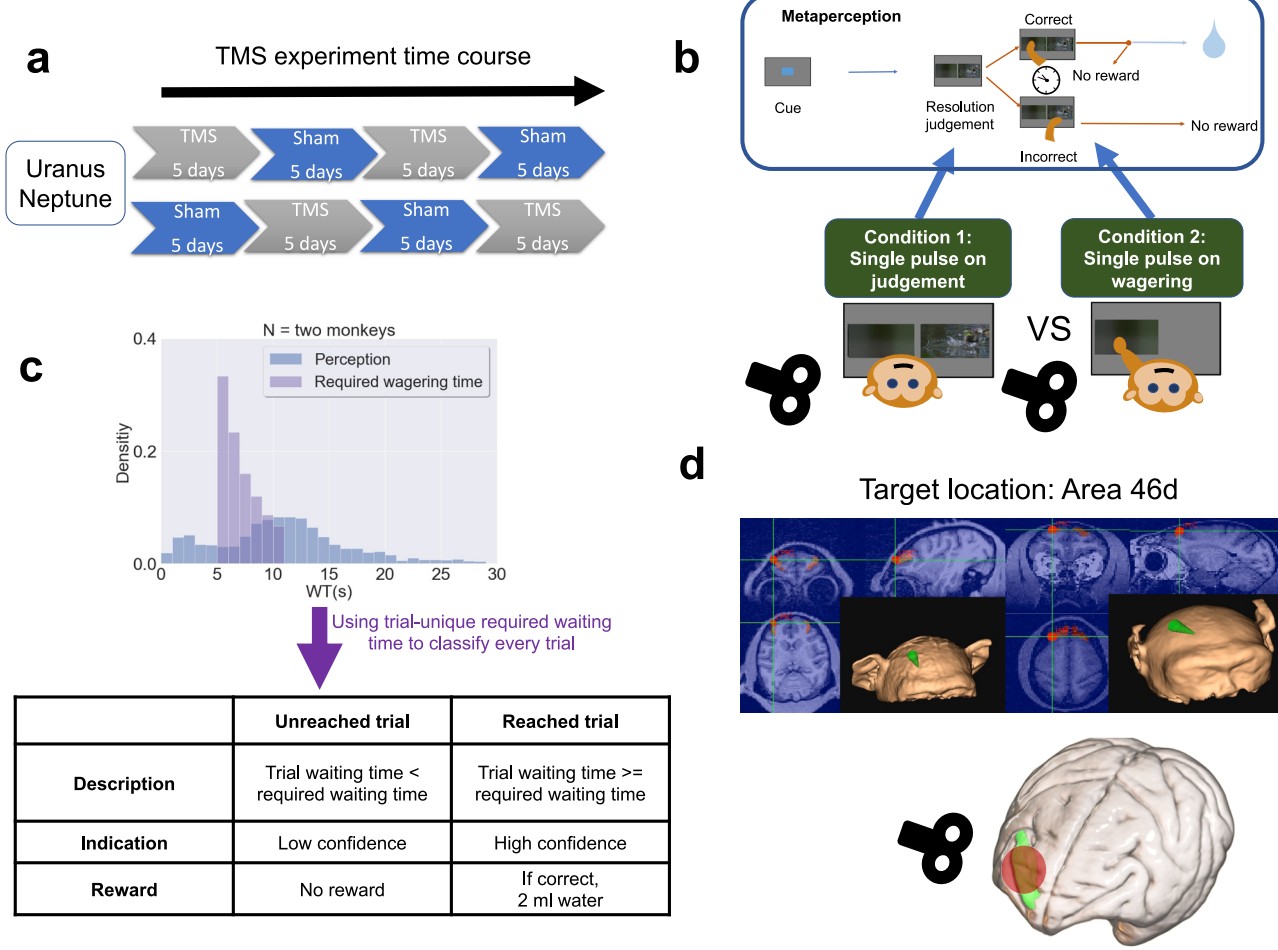

**Fig. 7 Temporal structure of the TMS experiment.** TMS experiment schedule with TMS-46d/sham conditions counterbalanced between monkeys (Uranus and Neptune) (**a**). Perceptual judgement task with temporal wagering. Each trial consisted of a starting (blue) cue, a delay lasting 1~6 s, and two simultaneously presented pictures. The monkeys needed to choose the picture with lower resolution (or higher resolution, counterbalanced across monkeys) by holding their hand on the touchscreen. The waiting process was initiated as soon as they laid their hand on the picture. Their confidence in the decision was measured by temporal wagering; that is, they could wait for a reward if they were confident or opt out to abort the current trial. There were two TMS conditions, which differed in the timing of stimulation. In each trial, the monkeys received a single TMS pulse either immediately after the onset of the picture stimulus (on-judgement phase) or 100 ms after they made their perceptual decision (on-wagering phase) (**b**). The required WT distribution and the actual WT distribution (only catch trials and incorrect trials) with WT bin size set to 1s. The table depicts the classification of low-confidence trials (unreached trials) and high-confidence trials (reached trials) (**c**). An illustration of the TMS site, as indicated by the green arrows. Bottom: The green area indicates BA46d on a rendering of a macaque brain; the red disc indicates the target area (**d**).

*Requirements for reward delivery and post-decision confidence measured by wagered time (WT).* Our study measured monkeys' confidence via a post-decision, time-based wagering paradigm. Following a monkey's perceptual or mnemonic decision, the animal needed to continue pressing the target (instead of merely tapping and releasing) to initiate a waiting process. The monkey would receive a reward (2 ml water) if it chose the correct picture *and* waited until the required WT set for that trial. The required WT for each trial was drawn from an exponential distribution with a decay constant equal to 1.5[18], and it differed from trial to trial, ranging from 5250 ms to 11,250 ms (with a new value selected every 500 ms) (Fig. 7c). We did not impose additional punishment measures such as a blank screen, considering that the WT itself served as an effective means of metacognitive feedback. The time duration that animals were willing to invest in each trial for a potential reward provided us with a quantitative measure of their trialwise decision confidence. We included catch trials (approximately 20% of correct trials) to reflect the maximum amount of wagered time, similar to a previous study[18]. In catch trials, we delivered the liquid reward after the monkeys released their hand off the screen.

*Training.* The preliminary training consisted of three main stages. First, we trained naïve monkeys to perform the perception and memory tasks separately. Note that the perceptual and mnemonic tasks require only brief touches as responses; thus, we avoided any preliminary training in confidence expression (no sustained contact required). Second, we introduced the requirement of sustained contact with the touchscreen for reward delivery: monkeys were trained to place their hand onto the screen and subsequently obtain a water reward with a single discrimination task

(choosing between a white rectangle and a yellow rectangle). The monkeys learned to keep their hand on the target for 3 s in this stage. Third, we introduced a contingency of random WTs, in which the maximum WT gradually increased from 5 s to 12 s. Catch trials were introduced in this stage. By the time of the experiments proper, we had the monkeys combine the perception and memory tasks with the sustained-contact wagering requirement from its outset.

**Data analysis.** In total, we registered 4624 trials for the TMS experiment and 16,177 trials for the domain-comparison experiment. Trials with RT longer than 10 s (6.3%) or shorter than 0.2 s (4.1%) were discarded from analysis in the domain-comparison experiment. We limited our WT-related analysis to trials with WT < 30 s (99.7% and 98.5% of trials were included in the TMS and domain-comparison experiments, respectively).

*Meta-index with hierarchical Bayesian estimation (hierarchical model meta-d′/d′).* Here, we calculated meta-d′/d′, a metric for estimating metacognitive efficiency (the level of metacognition given a level of performance or signal processing capacity) with a hierarchical Bayesian estimation method, which can avoid edge-correction confounds and enhance statistical power[75]. Meta-d′ is a measure of metacognitive accuracy from the empirical Type II receiver operating characteristic curve, which reflects the link between the subject's confidence and performance. To ensure that our results were not due to any idiosyncratic violation of the parametric assumptions of SDT, we additionally calculated a contingency index of preference

for the optimal choice[29,50] using the number of trials classified in each case [$n$(case)]:

$$Phi\ coefficient\ (\Phi) = \frac{n(Correct\ High) \times n(Incorrect\ Low) - n(Correct\ Low) \times n(Incorrect\ High)}{\sqrt{n(Correct) \times n(Incorrect) \times n(High) \times n(Low)}} \quad (1)$$

*Classification of high- and low-confidence trials.* In order to compute meta-d′/d′ and the phi coefficient, it is necessary to find the distribution of four trial types: high confidence/correct, low confidence/incorrect, low confidence/correct, and low confidence/incorrect. We used the trial-specific required waiting time to classify every trial as high confidence or low confidence, similar to the way confidence is binarized into high and low in human studies[4,6,76]. Specifically, we designated the unreached trials (where the actual wagered time was shorter than the required wagered time, in which case the monkeys would not receive a reward) as low-confidence trials. We designated the reached trials (where the actual wagered time was longer than or equal to the required wagered time, in which case the monkeys would receive a reward if the response was correct) as high confidence trials. We obtained one meta-d′/d′ and one phi coefficient per monkey per daily session.

*Logistic regression to probe the response-tracking precision of wagered time (WT).* By running logistic regression to capture how well WT might align with accuracy at the trial level, we tested for differences between tasks in the domain-comparison experiment (memory/perception) and between the two conditions in the TMS experiment (TMS-sham/46d) in terms of their respective WT response-tracking precision. We used only catch and incorrect trials in the logistic regression analysis.

In the domain-comparison experiment, we fit the percentage of correct responses with a logistic function containing WT, task (memory/perception), and the cross-product of WT as items and task to a logistic function:

$$P(correct) = \frac{1}{1 + e^{-(\beta_1 \times WT + \beta_2 \times task + \beta_3 \times WT \times task)}} \quad (2)$$

where $\beta_1$ reflects the response-tracking precision of WT, $\beta_2$ reflects the difference in accuracy between two tasks, and $\beta_3$ reflects the difference in WT response-tracking precision between tasks (memory/perception).

In the TMS experiment, we fit the percentage of correct responses to a logistic function with WT, TMS condition (TMS-46d/sham), and the cross-product of WT and TMS as terms:

$$P(correct) = \frac{1}{1 + e^{-(\beta_1 \times WT + \beta_2 \times TMS + \beta_3 \times WT \times TMS)}}, \quad (3)$$

where $\beta_1$ reflects the response-tracking precision of WT, $\beta_2$ reflects the difference in accuracy between two tasks, and $\beta_3$ reflects the difference in WT response-tracking precision between TMS conditions (TMS-46d/sham).

*Generalized linear models (GLMs).* We used GLMs to examine how WTs might vary as a function of task difficulty levels (see trial-difficulty psychometric curves in Fig. 4c–f). We used the *Enter* method to include several variables and their cross-products as items in the GLMs:

$$E(Y) = g^{-1}(X\beta), \quad (4)$$

where the dependent variable $Y$ is WT, $\beta$ is an unknown parameter to be estimated, and $g$ is a Gaussian estimated function. The independent variables $X$ are resolution difference, a binary regressor indicating correctness, a binary regressor indicating TMS modulation (TMS-46d/TMS-sham), a binary regressor indicating TMS phase (on-judgement/on-wagering), and their cross-product items.

*Domain-generality index (DGI) & pairwise correlation assessing metacognitive efficiency similarity of two tasks across and within subjects.* The DGI quantifies the similarity between scores in each domain[4] as follows:

$$DGI\ (domain - generality\ index) = \left| M_P - M_M \right|, \quad (5)$$

where $M_P$ is the perceptual H-model *meta-d′/d′* and $M_M$ is the memory H-model *meta-d′/d′*. Lower DGI scores indicate greater similarity in metacognitive efficiency between domains (DGI = 0 indicates identical scores).

In terms of pairwise correlation matrices, we built a matrix in which each entry E (task, monkey) represents the meta-efficiency correlation between a particular monkey and a particular task over a period of 20 days. For example, (M_Mars, P_Mars) represents the correlation between the meta-efficiency of the 20-day memory task and the 20-day perception task for Mars (Fig. 6f). An average-linkage clustering method[77] was employed to compute the minimum pairwise distance and generate a hierarchical cluster. These allowed us to test whether the within-task similarity exceeded the within-subjects similarity of two domains. We note that the DGI analyses were performed based on daily sessional data rather than within-session data. Therefore, the staircase procedure used in the perception task (within-session) should not influence our main results.

**Apparatus.** The training and testing were conducted in an automated test apparatus. The subject sat in a Plexiglas monkey chair (29.4 cm × 30.8 cm × 55 cm) fixed in position in front of an 18.5-inch capacitive touch-sensitive screen (Guangzhou TouchWo Co., Ltd, China) on which the stimuli could be displayed, and the monkeys were allowed to move their hands to press and hold the target. An automated water delivery reward system (5-RLD-D1, Crist Instrument Co., Inc, U.S.) delivered water through a tube positioned just beneath the mouth of the monkeys in response to the correct choices made by the subject. Apart from the backdrop lighting from the touch screen, the entire chair was placed in a dark experimental cubicle. The stimulus display and data collection were controlled by Python programs on a computer with millisecond precision. An infrared camera and a video recording system (EZVIZ-C2C, Hangzhou Ezviz Network Co., Ltd, China) were used to monitor the subjects.

**Material.** Documentary films on wild animals were gathered from YouTube and bilibili, including Monkey Kingdom (Disney), Monkey Planet (Episode 1–3; BBC), Monkey Thieves (http://natgeotv.com/asia/monkey-thieves), Monkeys: An Amazing Animal Family (https://skyvision.sky.com/programme/15753/monkeys--an-amazing-animal-family), Nature's Misfits (BBC), Planet Earth (Episode 1–11; BBC), Big Cats (Episode 1–3; BBC), and Snow Monkey (PBS Nature). In total, we collected 36 hours of video. We used Video Studio X8 (Core Corporation) to split the film into smaller clips (2 s each), and we used the CV2 package in Python to eliminate any blank frames. We chose 800 2-s clips that did not contain snakes, blank screens, or altered components such as typefaces as the video pool. We extracted 1600 still frames (two frames per video: 10th and 10th last frames) from these 800 clips.

**Statistics and reproducibility.** Data were analyzed using Python 3.6, Matlab R2019b, JASP 0.12. Significance was defined as $P < 0.05$. Mann–Whitney U tests was used to assess correct and incorrect trial WT differences. Logistic regression was used to probe the response-tracking precision of WT. Hierarchical model meta-d′/d′ was used to measure the metacognitive efficiency. GLMs were used to examine how WTs might vary as a function of task difficulty levels. Four macaque monkeys were involved. We adhered to the reduction principle in 3Rs and appropriately designed the experiments in such a way to minimize the number of animals used per experiment. No subjects were excluded. A small proportion of trials that did not meet reaction time requirement were not analyzed. It is a within-subjects design; we replicated the main findings across the monkeys. We randomized experimental conditions between subjects (e.g., order of TMS stimulation 46d vs. sham).

**Reporting summary.** Further information on research design is available in the Nature Research Reporting Summary linked to this article.

## Data availability

The source data underlying the main figures are provided as Supplementary Data 1. The raw data is available upon request.

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

## Acknowledgements

This research received support from the National Natural Science Foundation of China, grant number: 32071060, Science and Technology Commission of Shanghai Municipality, grant number: 201409002800, Open Research Fund of the State Key Laboratory of Cognitive Neuroscience and Learning (Beijing Normal University), internal funding from School of Psychology and Cognitive Science (East China Normal University), and Jiangsu Qinglan Talent Program Award (S.C.K.). Shanghai Municipal Education Commission— Gaofeng Clinical Medicine Grant Support, grant number: 20191836 (Y.T.). We thank Yong-di Zhou for his advice on NHP research; Makoto Kusunoki for implanting the headposts; and Lei Wang, Shuzhen Zuo, Angie Xie, Aihua Chen, Hakwan Lau, and Alicia Izquierdo for their input in the preparation of the manuscript.

## Author contributions

Y.C. and S.C.K. conceived the study. Y.C., Z.J., and C.Z. conducted the experiments. J.W. and Y.T. provided the TMS equipment. Y.C. performed the data analysis. All authors contributed to the interpretation of the results. Y.C. wrote the first draft of the manuscript with input from Z.J., C.Z., H.W., and Y.T. Y.C. and S.C.K. produced the final manuscript. S.C.K. supervised the project.

## Competing interests

The authors declare no competing interests.
