## [Peer Review File · Communications Biology]

Reviewers' comments:

Reviewer #1 (Remarks to the Author):

The manuscript entitled "Time-sensitive prefrontal involvement in associating confidence with task performance illustrates metacognitive introspection in monkeys" by Yudian Cai and colleagues addressed two findings of metacognitive abilities in monkeys. Using a post-decisional time wagering (WT) paradigm, the monkeys' metacognitive abilities could be assessed at the trial-by-trial basis. The authors then used online single-pulse TMS to perturb the right dIPFC region of monkeys ($n = 2$) and checked the metacognitive ability changes. Importantly, they administered the TMS pulses on the decision-making phase and the post-decisional wagering phase, respectively. They nicely analyzed the changes of WTs and their associations with the decision outcomes (correct or incorrect), and found that only online-decisional TMS perturbations, but not post-decisional TMS perturbations, affected the monkeys' metacognitive abilities. On the other hand, they also compared the metacognitive abilities in perception and memory in monkeys ($n = 4$). Their results showed that the metacognitive abilities are uncorrelated with each other, suggesting that the monkeys' metacognitive abilities in the two different domain tasks were separate. This study is well-designed and analyzed, the results are provoking, particularly applying the human paradigm to monkeys. Although this study is quite interesting, there are several main concerns that have not been addressed in the current version.

Major points:

1. The two findings are almost independent. It is much expected that the two domain tasks should be conducted in the TMS experiment, and seeing whether the perturbations should cause similar or distinct effects on the metacognitive abilities in the two tasks. However, the TMS experiments only conducted in the perception task.
2. It seems not straightforward to use trial-unique required waiting time, but not fixed required waiting time. Since the subjects did not know what should be the necessary waiting time for each trial, there exist a trade-off to wait a longer time for correct choice or to quit that one immediately for another opportunity of possible shorter waiting time. This paradigm needs quite a complex strategy to optimally respond, even for humans.
3. As the performance accuracy was controlled by a staircase procedure, it seems not reasonable to compare the performance accuracy between the different TMS perturbations. Even the TMS perturbations would affect performance, the accuracy should be not much changed. It is better to compare the task difficulty. Further, the overall performance accuracy controlled by the 4up-1down procedure might be two higher to give rise to ceiling effect about confidence.
4. Although the authors also used the metric of ϕ to assess the monkey's metacognitive abilities, the main figures only reported H-model $\text{meta-}d'/d'$. It might be better to show the reliability of the metrics.
5. The authors did not address the reason why the TMS target region was the right dIPFC. The cited literature is limited to human studies. Given differences between humans and monkeys, it seems that the selection of this target region is quite unsure. Hence, it is much expected that some control target region should be included to compare. Further, it remains unclear why TMS was only administered on two monkeys, but not four monkeys.
6. The TMS administrations were block-wise (5 days stimulation and 5 days sham) and the stimulations on the two phases were separate in two different sessions (20 days each session). Given the day-by-day variance, it is much expected to administer the two stimulations on the same block at the trial-by-trial basis.
7. The selection of TMS timings is not clear. Given that the single-pulse TMS effect should be quite weak, it remains unclear how the right dIPFC selectively affected metacognition at the very early stage of decision-making, in which even decision-making per se was not formed.
8. In the comparison between the two tasks, it looks odds to see that the performance accuracies were correlated, but the metacognitive abilities were not. As the performance accuracy was controlled in the perception task, but not possible in the memory task.
9. It also looks unusual to see high correlations of metacognitive abilities for the same task between subjects, but less correlations for the two tasks within subjects. Individual differences in metacognitive abilities considerably vary, at least, in humans. This unusual consistency between subjects might be caused by the consistency of performance in the same task, but differences between the two tasks (see point #8).

10. The definition of DGI using the absolute value of difference might be inappropriate. As the absolute value of random differences might be significant. It might be better to use the signed difference.

Minor points:

1. In Fig 2, the dots representing individual performance did not match the mean values.
2. P278-282. The statistical tests should be addressed in comparison between subjects and within subjects
3. The interpretation of relationship between RTs and WTs should be cautious. Please check whether the performance accuracy should be correlated with RTs.
4. Please also check whether WTs should be trial-by-trial correlated with the task difficulties.

Reviewer #2 (Remarks to the Author):

Cai and colleagues clearly demonstrated that area 46d played a critical role in supporting metacognition independent of task performance. The behavioural and brain stimulation paradigm that they developed in the present study is rigid, and the observations are clear and robust. It is especially great that they proved the existence of general metacognition ability and neural substrate in macaque monkeys using both memory and perception tasks. The reviewer believes the study will be a milestone to understanding the neural mechanism of metacognition in the prefrontal cortex.

1) How did the authors determine the location of the stimulation site? The authors should provide information about the stimulation site (MNI coordinates of stimulation sites, the methods to target the area, etc.). What is the authors' rationale to target the area 46d in the experiment? Are there any reason why the authors stimulated only on the right side not on both hemispheres? Please explain these points more in detail.

2) Early meta-perception study of monkeys by Middlebrooks and Sommer (2012) reported that the neuronal activity at the dorsolateral prefrontal cortex did not encode metacognition at the population level. Instead, they reported that the supplementary eye field was active in correlation with the monkey's metacognitive judgments. How do the authors consider the relationship between the findings in the current study and the previous study? Please discuss.

3) In Figure 2, I suppose that the trace describes the average performance across the four monkeys. I wonder why in the memory condition (panels C and D) the trace is generally smaller than the four individual monkeys' data; a similar effect is not observed in the perception condition. Please explain.

4) The reviewer is interested in the results shown in Figure 4C. As the authors discuss, it is possible that monkeys started relying on RT as an associative cue after having received TMS on area 46d. I am curious whether this strategy switch contributed to compensating the metacognitive performance impairments or not. Was the negative correlation between RT and WT common for both correct and incorrect trials? Did RT predict the performance of the first-order task? These questions will not influence the main conclusion of the paper. But they will help to understand how the judgment-related information processed at area 46d is used for wagering.

5) It is great that the authors tested metacognitive ability across different tasks in the same monkeys. Stimulation of area 46d by TMS induced impairments of metacognitive ability (meta-d') for both perception and memory tasks, whereas domain-general index demonstrated that within-task similarity of metacognitive efficiency was stronger than the within-subjects similarity. These two observations seem to be a bit competing. How do authors consider the role of the same area 46d for metacognition of memory and perception? Which process does the area 46d contribute to? Domain-specific or domain-general metacognition? One possibility is that the locus of domain-specific metacognition for perception and memory is close to each other around the area 46d and TMS affects activities in both areas. Please discuss these points.

Reviewer #1(Remarks to the Author):

The manuscript entitled “Time-sensitive prefrontal involvement in associating confidence with task performance illustrates metacognitive introspection in monkeys” by Yudian Cai and colleagues addressed two findings of metacognitive abilities in monkeys. Using a post-decisional time wagering (WT) paradigm, the monkeys’ metacognitive abilities could be assessed at the trial-by-trial basis. The authors then used online single-pulse TMS to perturb the right dlPFC region of monkeys (n = 2) and checked the metacognitive ability changes. Importantly, they administered the TMS pulses on the decision-making phase and the post-decisional wagering phase, respectively. They nicely analyzed the changes of WTs and their associations with the decision outcomes (correct or incorrect), and found that only online-decisional TMS perturbations, but not post-decisional TMS perturbations, affected the monkeys’ metacognitive abilities. On the other hand, they also compared the metacognitive abilities in perception and memory in monkeys (n = 4). Their results showed that the metacognitive abilities are uncorrelated with each other, suggesting that the monkeys’ metacognitive abilities in the two different domain tasks were separate. This study is well-designed and analyzed, the results are provoking, particularly applying the human paradigm to monkeys. Although this study is quite interesting, there are several main concerns that have not been addressed in the current version.

RESPONSE: We thank the reviewer for the constructive suggestions. We have re-examined the data more thoroughly, and detailed new results as follows based on the suggested methods. The manuscript has been revised substantially in light of these new results and discussion.

Major points:

1. The two findings are almost independent. It is much expected that the two domain tasks should be conducted in the TMS experiment, and seeing whether the perturbations should cause similar or distinct effects on the metacognitive abilities in the two tasks. However, the TMS experiments only conducted in the perception task.

RESPONSE: Thank you very much for this important consideration. We believe that in the framework of perception, our TMS experiment provides a demonstration of introspection and time-sensitive prefrontal involvement. However, indeed, the time sensitive role of prefrontal cortex in both perceptual and mnemonic tasks is definitely an interesting and important topic that needs to be addressed in the future. We therefore added a discussion of this issue in the revised manuscript (lines 413-431; see also our response to reviewer 2, comment #5).

Recent studies showed that BOLD signals around BA46d in dlPFC in the macaque brain are associated with metamemory (Miyamoto et al., 2017; Miyamoto et al., 2018), whereas our current results showed a causal role of BA46d in dlPFC in meta-perception. This led us to suggest that BA46d might have a domain-general role in metacognition. We also found that the TMS pulse affected on-judgement phase but not on-wagering phase, indicating BA46d is especially functionally related to processes underlying evidence accumulation during decision. We postulate that BA46d accumulates domain-general evidence and relates the information to some downstream domain-specific areas such as the SEF.

2. It seems not straightforward to use trial-unique required waiting time, but not fixed required waiting time. Since the subjects did not know what should be the necessary waiting time for each trial, there exist a trade-off to wait a longer time

for correct choice or to quit that one immediately for another opportunity of possible shorter waiting time. This paradigm needs quite a complex strategy to optimally respond, even for humans.

RESPONSE: We appreciate the reviewer's suggestion on the possibility of using a fixed required waiting time. Fixed required waiting time could be more straightforward and easier to implement in the experiment. However, to minimize the likelihood that the animals might develop a strategy to wait for the optimal waiting time (fixed waiting time) irrespective of the response outcome, we decided to adhere to the standard temporal wagering paradigms used in rodent studies (Lak et al., 2014; Stolyarova, et al. 2019; Masset et al., 2020). As shown in the main results (e.g., Fig. 3d), a trial-unique required waiting time approach seems to be sensitive to reflect animals' judgement of their performance.

3. As the performance accuracy was controlled by a staircase procedure, it seems not reasonable to compare the performance accuracy between the different TMS perturbations. Even the TMS perturbations would affect performance, the accuracy should be not much changed. It is better to compare the task difficulty. Further, the overall performance accuracy controlled by the 4up-1down procedure might be too high to give rise to ceiling effect about confidence.

RESPONSE: To address this concern, we compared the distributional differences between the TMS conditions and we did not find significant differences in task difficulty between TMS-46d and TMS-sham conditions in either on-judgement phase (Mann–Whitney U test results: Uranus, $p = 0.074$; Neptune, $p = 0.804$) or on-wagering phase (Mann–Whitney U test results: Uranus, $p = 0.158$; Neptune, $p = 0.635$) (lines 256-262 in the revised manuscript and revised Figure 5g-h).

In terms of accuracy, we managed to keep the overall performance accuracy in the range of 62.6 - 86.3% (mean: 81.7% \pm 3.6%), which is within a reasonable range compared to a recommended accuracy (cf. ~71% as discussed in Fleming et al. 2010; Fleming et al, 2012). We also believe that if the monkeys reached ceiling in accuracy, the metacognitive judgement shall be skewed, and the chance leading to differences between TMS-sham and TMS-46d conditions would be very negligible. We added reporting of the performance accuracy in the revised manuscript (lines 262-270).

4. Although the authors also used the metric of phi to assess the monkey's metacognitive abilities, the main figures only reported H-model meta-d'/d'. It might be better to show the reliability of the metrics.

RESPONSE: We concur with the reviewer's suggestion on reporting the reliability of the metrics and we have now reported that the two metrics were highly correlated (Pearson correlation: experiment domain-comparison, $r = 0.7916$, $p < 0.001$; experiment TMS, $r = 0.7415$, $p < 0.001$), confirming reliability between Phi and Hmodel-meta d'/d'. We have added these statistical results in the revised manuscript (lines 117-120) and in a revised Figure 2e-f.

5. The authors did not address the reason why the TMS target region was the right dIPFC. The cited literature is limited to human studies. Given differences between humans and monkeys, it seems that the selection of this target region is quite unsure. Hence, it is much expected that some control target region should be included to compare. Further, it remains unclear why TMS was only administered on two monkeys, but not four monkeys.

RESPONSE: We thank the reviewer for this comment. In the revised manuscript (lines 58-62 & 547-550), we have now included a more balanced discussion by considering the NHP literature more fully. We added two important recent studies which showed that neural activations in dorsal PFC and anterior PFC in the macaque brain are associated with metacognition of mnemonic experienced object recognition (Miyamoto et al., 2017; Miyamoto et al., 2018). An earlier single neuron study also indicated the metacognitive involvement of dIPFC in monkeys (Middlebrook et al., 2012) (see also reviewer 2, comment #2). Another consideration regarding the choice of hemisphere is that the right hemisphere has been shown dominant for visual processing (Hellige, 1996). Given that our perceptual task is a highly visually demanding task and that in human TMS studies the manipulations were also conducted in the right dIPFC (Shekar et al., 2018; Rounis et al., 2010), we believe that the

right dlPFC could likely act as a key site for (perceptual) introspection in the macaques.

For control purposes, we opted for a “sham-condition” approach. By this, we rotated the coil 90 degrees over BA46d, thereby ensuring that the sound and vibration (by-products) of the stimulation were identical between the TMS-46d and TMS-sham conditions. Since we have head-posts implanted near the mid-line on the two monkeys, options for control sites (e.g., homologue for the human vertex) were very limited operationally. We added this justification in the revised manuscript (lines 526-531). We have also added that since only two of the monkeys were implanted with head-posts, which was a prerequisite for maintaining their heads steadily for the TMS stimulation, TMS experiment data was acquired only from these two monkeys (Uranus and Neptune) (lines 499-502).

6. The TMS administrations were block-wise (5 days stimulation and 5 days sham) and the stimulations on the two phases were separate in two different sessions (20 days each session). Given the day-by-day variance, it is much expected to administrate the two stimulations on the same block at the trial-by-trial basis.

RESPONSE: We thank the reviewer for the important consideration. While a trial-wise TMS design would reduce day-by-day variance, trial-wise TMS might likely cause strong inter-trial influences across consecutive trials. For example, it would be impossible to rule out the possibility that an “on-wagering” pulse would not linger and impact on the “on-judgement” phase in the subsequent trial. We thus decided to administer a block-wise TMS protocol. By having counterbalanced the testing order for the two factors (order of TMS-46d/sham and on-judgement/on-wagering conditions) within and across monkeys, this design would have minimized any factor-related and day-by-day variances.

7. The selection of TMS timings is not clear. Given that the single-pulse TMS effect should be quite weak, it remains unclear how the right dlPFC selectively affected metacognition at the very early stage of decision-making, in which even decision-making per se was not formed.

RESPONSE: We also hold the opinion that the precise timing for decision-making is not entirely clear. Here, we made the assumption that meta-calculation could be an integral part of first-order decision making and noting that in a previous study (Shekhar & Rahnev, 2018) that TMS at 250ms would lead to deficits in confidence calculation, we therefore set the TMS timing as close to the stimuli onset as possible (that is, 100ms after stimuli-onset). Because the transient TMS evoked potential (TEP) as it may, we obtained a strong dissociation between meta-indices and accuracy, hinting a possibility that the dlPFC performs meta-calculation at a very early stage. We have refined our discussion to take these considerations into account (lines 372-378).

8. In the comparison between the two tasks, it looks odd to see that the performance accuracies were correlated, but the metacognitive abilities were not. As the performance accuracy was controlled in the perception task, but not possible in the memory task.

RESPONSE: The DGI analyses were performed based on daily sessional data rather than within-session data. Therefore, the staircase procedure used in the perception task (within-session) should not influence our main results. On this note, given the monkeys had been trained extensively for over 6 months on a variant of this memory task, they were already exceedingly stable in their memory performances across the period of testing. Altogether, it is unlikely that this procedural difference between tasks has affected the key findings. We have added this note in the Methods section (lines 483-485 & 664-667). In light of comment #9, we have scrutinized the data further by running a new analysis on the factor "accuracy" and found accuracy itself did not show any (task) domain-

specific effect.

9. It also looks unusual to see high correlations of metacognitive abilities for the same task between subjects, but less correlations for the two tasks within subjects. Individual differences in metacognitive abilities considerably vary, at least, in humans. This unusual consistency between subjects might be caused by the consistency of performance in the same task, but differences between the two tasks (see point #8).

RESPONSE: To directly address this concern, we performed the same clustering and random shuffle simulation on the factor “accuracy”. In contrast to “Hmodel-meta d'/d ”, accuracy itself did not show any domain-specific effect. This indicates that the within-task consistency is dissociable for the two factors, “accuracy” vs. “meta”.

We also took the suggestion made in the reviewer’s minor comment #2. We calculated the pairwise standardized Euclidean distance of each vector pair (in total 28 vector pairs, each vector corresponding to each row in revised Figure 7j, each row containing 8 cells) and showed that there were no statistical differences in accuracy standardized Euclidean-distance across and within monkeys (Mann–Whitney U test results: $p = 0.380$). These results are now reported in lines 307-325).

F

Example	Description
Within-monkey Pair: M_Mars, P_Mars	20 days correlation across domains: Panel G: Meta Panel I: Accuracy
Across-monkey Pair: M_Mars, M_Saturn	20 days correlation across monkeys: Panel H: Meta Panel J: Accuracy

M = Memory task; P = Perceptual task
 Monkeys = Mars and Saturn

10. The definition of DGI using the absolute value of difference might be inappropriate. As the absolute value of random differences might be significant. It might be better to use the signed difference.

RESPONSE: As per the reviewer's request, we performed Mann–Whitney U test on signed DGI and replicated the domain-specific effects (all four monkeys, $p < 0.001$; Mars, $p = 0.153$; Saturn, $p < 0.001$; Uranus, $p < 0.001$; Neptune, $p = 0.263$). We have added these new results in the revised manuscript (lines 308-311). In order to be compatible with the literature (Fleming et al., 2014; Morales et al., 2018), we maintained to use the absolute values for this analysis and have added these references in the revised manuscript (lines 307-308).

Minor points:

1. In Fig 2, the dots representing individual performance did not match the mean values.

RESPONSE: We corrected a plotting error and revised Figure 2.

2. P278-282. The statistical tests should be addressed in comparison between subjects and within subjects

RESPONSE: We calculated the pairwise standardized Euclidean distance of each vector pair (in total 28 vector pairs, each vector corresponding to each row in the revised Figure 7g-j, each row containing 8 cells) and found pairwise distance of Hmodel-*meta* d'/d' in within-tasks across monkeys are significantly shorter than across-tasks within monkeys (Mann–Whitney U test results: $p = 0.033$), but not for pairwise distance of accuracy (Mann–Whitney U test results: $p = 0.380$). These results are now reported in lines 316-325.

3. The interpretation of relationship between RTs and WTs should be cautious. Please check whether the performance accuracy should be correlated with RTs.

RESPONSE: We found a strong negative correlation between accuracy and RT. Specifically, we showed RT was negatively correlated with accuracy in the domain-comparison experiment (perception, $r = -0.0819$, $p < 0.001$; memory, $r = -0.17535$, $p < 0.001$), and in both on-judgement (TMS-46d, $r = -0.0856$, $p = 0.0038$; Sham, $r = -0.1345$, $p < 0.001$) and on-wagering (TMS-46d, $r = -0.0983$, $p < 0.001$; Sham, $r = -0.1063$, $p < 0.001$) phase in the TMS experiment (see lines 217-225 and revised Figure 4d-f).

4. Please also check whether WTs should be trial-by-trial correlated with the task difficulties.

RESPONSE: We found a Pearson correlation between WT and task difficulty in the TMS experiment (for two monkeys: $r = -0.062$, $p = 0.010$; *Uranus*, $r = -0.0710$, $p = 0.046$; *Neptune*, $r = -0.108$, $p = 0.002$). These statistics are now added in the revised manuscript (lines 241-243).

Reference:

Fleming, S. M., Weil, R. S., Nagy, Z., Dolan, R. J. & Rees, G. Relating introspective accuracy to individual differences in brain structure. *Science* **329**, 1541-1543 (2010).

Fleming, S. M. & Dolan, R. J. The neural basis of metacognitive ability. *Phil. Trans. R. Soc. B* **367**, 1338–1349 (2012).

Hellige, Joseph B. Hemispheric asymmetry for visual information processing. *Acta Neurobiol Exp* **56**, 485-497 (1996).

Middlebrooks, P. G. & Sommer, M. A. Neuronal correlates of metacognition in

primate frontal cortex. *Neuron* **75**, 517-530 (2012).

Miyamoto, K. *et al.* Causal neural network of metamemory for retrospection in primates. *Science* **355**, 188-193 (2017).

Miyamoto, K., Setsuie, R., Osada, T. & Miyashita, Y. Reversible silencing of the frontopolar cortex selectively impairs metacognitive judgment on non-experience in primates. *Neuron* **97**, 980-989.e6 (2018).

Morales, J., Lau, H. & Fleming, S. M. Domain-general and domain-specific patterns of activity supporting metacognition in human prefrontal cortex. *J. Neurosci.* **38**, 3534-3546 (2018).

Rounis, E., Maniscalco, B., Rothwell, J. C., Passingham, R. E. & Lau, H. Theta-burst transcranial magnetic stimulation to the prefrontal cortex impairs metacognitive visual awareness. *Cogn. Neurosci.* **1**, 165-175 (2010).

Shekhar, M. & Rahnev, D. Distinguishing the roles of dorsolateral and anterior PFC in visual metacognition. *J. Neurosci.* **38**, 5078-5087 (2018).

Reviewer #2 (Remarks to the Author):

Cai and colleagues clearly demonstrated that area 46d played a critical role in supporting metacognition independent of task performance. The behavioural and brain stimulation paradigm that they developed in the present study is rigid, and the observations are clear and robust. It is especially great that they proved the existence of general metacognition ability and neural substrate in macaque monkeys using both memory and perception tasks. The reviewer believes the study will be a milestone to understanding the neural mechanism of metacognition in the prefrontal cortex.

RESPONSE: We thank the reviewer for their kind and constructive comments. In the following, we will provide detailed responses to each of them.

1. How did the authors determine the location of the stimulation site? The authors should provide information about the stimulation site (MNI coordinates of stimulation sites, the methods to target the area, etc.). What is the authors' rationale to target the area 46d in the experiment? Are there any reason why the authors stimulated only on the right side not on both hemispheres? Please explain these points more in detail.

RESPONSE: Based on a review of the literature in both human and NHP studies, we believe that the right dlPFC could likely act as a key site for (perceptual) introspection in the macaques. For example, the monkey dlPFC is involved in metacognitive decisions (Middlebrook et al., 2012) and that neural activations in dorsal PFC and anterior PFC in the macaque brain are associated with metacognition of mnemonic experienced object recognition (Miyamoto et al., 2017; Miyamoto et al., 2018). Moreover, the right hemisphere has been shown dominant for visual processing (Hellige, 1996). Given that our perceptual task is a highly visually demanding task and that in human TMS studies the manipulations were usually conducted in the right dlPFC (Shekar et al., 2018;

Rounis et al., 2010), we decided to choose the right dlPFC as our target site. We have added these details in the revised manuscript (lines 58-62 & 547-550) (see also reviewer 1, comment #5). We have also a detailed description on the stimulation site in the revised manuscript (lines 536-542).

2. Early meta-perception study of monkeys by Middlebrooks and Sommer (2012) reported that the neuronal activity at the dorsolateral prefrontal cortex did not encode metacognition at the population level. Instead, they reported that the supplementary eye field was active in correlation with the monkey's metacognitive judgments. How do the authors consider the relationship between the findings in the current study and the previous study? Please discuss.

RESPONSE: We thank the reviewer for pointing us to consider the frontal metacognitive systems more globally. We accord that "it is likely that dlPFC and other regions such as the supplementary eye field (SEF) play different roles at different metacognitive stages. It is known that the supplementary eye field neurons encode metacognitive components in a meta-perception study (Middlebrook et al., 2012; Abzug et al., 2018). The authors suggested a possibility the metacognitive information is not readily stored in the SEF, but was generated elsewhere, and would be transmitted to the SEF for use when it is required. To put this evidence into perspective, we are inclined to postulate that BA46d accumulates domain-general evidence and relates the information to some downstream domain-specific areas such as the SEF." We add these views on the frontal metacognitive systems into the revised manuscript (lines 419-431).

3. In Figure 2, I suppose that the trace describes the average performance across the four monkeys. I wonder why in the memory condition (panels C and D) the trace is generally smaller than the four individual monkeys' data; a similar effect is not observed in the perception condition. Please explain.

RESPONSE: We corrected a plotting error and revised Figure 2 in the revision.

4. The reviewer is interested in the results shown in Figure 4C. As the authors discuss, it is possible that monkeys started relying on RT as an associative cue after having received TMS on area 46d. I am curious whether this strategy switch contributed to compensating the metacognitive performance impairments or not. Was the negative correlation between RT and WT common for both correct and incorrect trials? Did RT predict the performance of the first-order task? These questions will not influence the main conclusion of the paper. But they will help to understand how the judgment-related information processed at area 46d is used for wagering.

RESPONSE: As per the reviewer's request, we performed correlation analysis and found a strong negative correlation between accuracy and RT. Specifically, we showed RT was negatively correlated with accuracy in the domain-comparison experiment (perception, $r = -0.0819$, $p < 0.001$; memory, $r = -0.17535$, $p < 0.001$), and in both on-judgement (TMS-46d, $r = -0.0856$, $p = 0.0038$; Sham, $r = -0.1345$, $p < 0.001$) and on-wagering (TMS-46d, $r = -0.0983$, $p < 0.001$; Sham, $r = -0.1063$, $p < 0.001$) phase in the TMS experiment (see lines 217-223 and revised Figure 4d-f).

We also found a negative correlation in TMS-46d condition in correct trials ($r = -0.266$, $p < 0.001$), and a negative correlation tendency in incorrect trials ($r = -$

0.1064, $p = 0.1336$) (see line 223-225 in the revised manuscript and revised Figure 4g-h).

5. It is great that the authors tested metacognitive ability across different tasks in the same monkeys. Stimulation of area 46d by TMS induced impairments of metacognitive ability (meta-d') for both perception and memory tasks, whereas domain-general index demonstrated that within-task similarity of metacognitive efficiency was stronger than the within-subjects similarity. These two observations seem to be a bit competing. How do authors consider the role of the same area 46d for metacognition of memory and perception? Which process does the area 46d contribute to? Domain-specific or domain-general metacognition? One possibility is that the locus of domain-specific metacognition for perception and memory is close to each other around the area 46d and TMS affects activities in both areas. Please discuss these points.

RESPONSE: We cannot agree more with you on the importance of this issue. Recent studies showed that BOLD signals around BA46d in dIPFC in the macaque brain are associated with metamemory (Miyamoto et al., 2017; Miyamoto et al., 2018), whereas our current results showed a causal role of BA46d in dIPFC in meta-perception. This led us to suggest that BA46d might have a domain-general role in metacognition. We also found that the TMS pulse affected on-judgement phase but not on-wagering phase (see our response to

reviewer's comment #2), indicating BA46d is especially functionally related to processes underlying evidence accumulation during decision. We postulate that BA46d accumulates domain-general evidence and relates the information to some downstream domain-specific areas. However, indeed, in the current study, it remains possible that we could not precisely demarcate the respective loci for domain-specific metacognition for perception vs. memory since our TMS might have affected a relatively large portion of the dlPFC. We have now added this discussion in the revised manuscript (lines 413-431).

Reference:

Abzug, Zachary M., and Marc A. Sommer. "Neuronal correlates of serial decision-making in the supplementary eye field." *J. Neurosci.* **38**, 7280-7292 (2018).

Hellige, Joseph B. "Hemispheric asymmetry for visual information processing." *Acta Neurobiol Exp* 56, 485-497 (1996).

Middlebrooks, P. G. & Sommer, M. A. Neuronal correlates of metacognition in primate frontal cortex. *Neuron* **75**, 517-530 (2012).

Miyamoto, K. *et al.* Causal neural network of metamemory for retrospection in primates. *Science* **355**, 188-193 (2017).

Miyamoto, K., Setsuie, R., Osada, T. & Miyashita, Y. Reversible silencing of the frontopolar cortex selectively impairs metacognitive judgment on non-experience in primates. *Neuron* **97**, 980-989.e6 (2018).

Rounis, E., Maniscalco, B., Rothwell, J. C., Passingham, R. E. & Lau, H. Theta-burst transcranial magnetic stimulation to the prefrontal cortex impairs metacognitive visual awareness. *Cogn. Neurosci.* **1**, 165-175 (2010).

Shekhar, M. & Rahnev, D. Distinguishing the roles of dorsolateral and anterior PFC in visual metacognition. *J. Neurosci.* **38**, 5078-5087 (2018).

REVIEWERS' COMMENTS:

Reviewer #2 (Remarks to the Author):

The authors responded fully to the reviewer's comments and now the reviewer is almost satisfied. But the reviewer thinks that the authors to be more careful about the sentence they added to the Discussion in response to the reviewer's comment 2.

In the current manuscript, they discuss that "the authors suggested a possibility the metacognitive information is not readily stored in the SEF, but was generated elsewhere, and would be transmitted to the SEF for use when it is required." Surely, the authors revealed that BA46d contributes to perceptual metacognition. But they did not directly test/prove that metacognitive information is not generated in the SEF nor transmitted to the SEF for use when it is required in the present study.

Reviewer #3 (Remarks to the Author):

Review of Time-sensitive prefrontal involvement in associating confidence with task performance illustrates metacognitive introspection in monkeys, by Cai and co-authors.

I am a 3rd reviewer of this manuscript and did not participate in previous review rounds. Cai et al., performed a meta-cognition experiment in monkeys, using a perceptual and a temporal-memory metacognition task and they targeting area 46 with TMS. They employed TMS both during decision and wagering phases of the task trials. They showed that this dLPFC area is involved in metacognition independent of performance, but the effects are task -dependent. They also conclude that area 46 accumulates domain-general evidence. In general, I find this is a very carefully conducted study, the data are well-analyzed and the writing is crisp. The results add significantly to the recent landmark papers on metacognition in nonhuman primates of Sommer's and Myiashita's research groups. The authors were able to address all the concerns of the previous reviewer and I do not have any additional major comment and recommend the manuscript for publication.

I only have two minor issues that may be considered however:

- 1) The Hellige 1996 manuscript is taken as evidence for a right hemispheric dominance for visual processing -triggered by a reviewer's comment. There is, however, very little, in my view if no, conclusive evidence for asymmetry of visual processing in nonhuman primates. I suggest to remove this argument, as it is not critical at all.
- 2) I suggest to include a short task description in the beginning of the result section. It is very weird to start describing effects, without any reference to the tasks itself. Just referring to the methods section is a somewhat odd. This can be done in 1-2 sentences.

Reviewer #2(Remarks to the Author):

The authors responded fully to the reviewer's comments and now the reviewer is almost satisfied.

But the reviewer thinks that the authors to be more careful about the sentence they added to the Discussion in response to the reviewer's comment 2.

In the current manuscript, they discuss that "the authors suggested a possibility the metacognitive information is not readily stored in the SEF, but was generated elsewhere, and would be transmitted to the SEF for use when it is required." Surely, the authors revealed that BA46d contributes to perceptual metacognition. But they did not directly test/prove that metacognitive information is not generated in the SEF nor transmitted to the SEF for use when it is required in the present study.

RESPONSE: We agree with the reviewer's comment. We have now deleted that statement in question in the revised manuscript.

Reviewer #2 (Remarks to the Author):

Review of Time-sensitive prefrontal involvement in associating confidence with task performance illustrates metacognitive introspection in monkeys, by Cai and co-authors.

I am a 3rd reviewer of this manuscript and did not participate in previous review rounds. Cai et al., performed a meta-cognition experiment in monkeys, using a perceptual and a temporal-memory metacognition task and they targeting area 46 with TMS. They employed TMS both during decision and wagering phases of the task trials. They showed that this dLPFC area is involved in metacognition independent of performance, but the effects are task -dependent. They also conclude that area 46 accumulates domain-general evidence. In general, I find this is a very carefully conducted study, the data are well-analyzed and the writing is crisp. The results add significantly to the recent landmark papers on

metacognition in nonhuman primates of Sommer's and Myiashita's research groups. The authors were able to address all the concerns of the previous reviewer and I do not have any additional major comment and recommend the manuscript for publication.

I only have two minor issues that may be considered however:

1) The Hellige 1996 manuscript is taken as evidence for a right hemispheric dominance for visual processing -triggered by a reviewer's comment. There is, however, very little, in my view if no, conclusive evidence for asymmetry of visual processing in nonhuman primates. I suggest to remove this argument, as it is not critical at all.

RESPONSE: We agree with the reviewer's comment. We have now deleted that statement in question in the revised manuscript.

2) I suggest to include a short task description in the beginning of the result section. It is very weird to start describing effects, without any reference to the tasks itself. Just referring to the methods section is a somewhat odd. This can be done in 1-2 sentences.

RESPONSE: We thank the reviewer for this suggestion. At the beginning of the methods section, we have now added two brief sentences to introduce the tasks that the monkeys have performed: "In the following, we will report results obtained from four monkeys who participated in two distinct tasks tapping into two metacognitive domains. Most critically, we measured the animal's trial-wise confidence level using a time-wagering paradigm." (Line 103-105)